# HourVideo: 1-Hour Video-Language Understanding

**Keshigeyan Chandrasegaran   Agrim Gupta   Lea M. Hadzic   Taran Kota   Jimming He**
**Cristobal Eyzaguirre   Zane Durante   Manling Li   Jiajun Wu   Li Fei-Fei**

hourvideo.stanford.edu

Stanford University

## Abstract

We present **HourVideo**, a benchmark dataset for hour-long video-language understanding. Our dataset consists of a novel task suite comprising summarization, perception (*recall, tracking*), visual reasoning (*spatial, temporal, predictive, causal, counterfactual*), and navigation (*room-to-room, object retrieval*) tasks. HourVideo includes 500 manually curated egocentric videos from the Ego4D dataset, spanning durations of 20 to 120 minutes, and features **12,976 high-quality, five-way multiple-choice questions**. Benchmarking results reveal that multimodal models, including GPT-4 and LLaVA-NeXT, achieve marginal improvements over random chance. In stark contrast, human experts significantly outperform the state-of-the-art long-context multimodal model, Gemini Pro 1.5 (85.0% vs. 37.3%), highlighting a substantial gap in multimodal capabilities. Our benchmark, evaluation toolkit, prompts, and documentation are available at hourvideo.stanford.edu.

## 1   Introduction

Humans demonstrate a remarkable ability to process visual stimuli over long time horizons, enabling them to perceive, plan and act in the real world. Consider the routine task of cooking a meal. This activity involves a continuous and adaptive visual process: identifying and using ingredients and tools, monitoring state changes of various dishes, and adjusting cooking duration/techniques based on visual cues such as color and texture. Such sustained visual processing is crucial to achieving the desired culinary outcomes. Naturally, endowing autonomous agents with this capability has been a long-standing goal in the field of Artificial Intelligence.

In recent years, large multimodal models [1–3] have emerged as a promising approach toward achieving this goal. Typically, these models are evaluated using multiple datasets that test capabilities such as object recognition [4, 5], image comprehension [6–8], and action recognition [9]. However, these benchmarks are often restricted to single images or short video clips, usually lasting from a few seconds to no more than three minutes [9–12]. While these benchmarks have spurred significant advancements, a deeper exploration into long-form video-language understanding is essential to develop multimodal systems that can form the basis for future autonomous agents and assistants.

A significant challenge in evaluating long-form video-language understanding capabilities is designing tasks that genuinely necessitate *long-term* comprehension, i.e., tasks that require long-range dependencies. Merely posing questions that can be answered by watching a brief segment of a lengthy video effectively reduces the task to a combination of temporal localization and short-clip understanding. Furthermore, while intriguing narrative inquiries can certainly be formulated for long-form videos such as television shows and films, it is imperative to ensure that the questions are not trivially answerable due to the vast prior knowledge encoded in modern large language models.

In this work, we introduce **HourVideo**—a benchmark dataset designed for long-form video-language understanding. To design tasks that require *long-term* comprehension, we first propose a novel task

---

Correspondence to {keshik,agrim}@stanford.edu

38th Conference on Neural Information Processing Systems (NeurIPS 2024) Track on Datasets and Benchmarks.

suite (Tab. 1), comprising **summarization**, **perception** (*recall, tracking*), **visual reasoning** (*spatial, temporal, predictive, causal, counterfactual*), and **navigation** (*room-to-room, object retrieval*) tasks. For each task, we manually create question prototypes designed to ensure that correctly answering them requires identification and synthesis of information across multiple temporal segments within the long-form videos. Guided by our task suite, we curated 500 egocentric videos from the Ego4D dataset [13]—covering 77 unique everyday activities and ranging from 20 to 120 minutes—to generate questions based on our prototypes. The combination of our comprehensive task suite and everyday mundane egocentric videos provides a robust framework to rigorously evaluate multimodal models' capabilities in understanding long-form videos. Finally, we developed a question-answer generation pipeline utilizing the expertise of trained human annotators (800+ hours of effort) and large language models (LLMs), resulting in a collection of 12,976 high-quality, five-way multiple-choice questions.

We comprehensively evaluate state-of-the-art multimodal models on HourVideo (Tab. 2, Fig. 4), including GPT-4V [2], Gemini 1.5 Pro [3], and LLaVA-NeXT [14] in a zero-shot setting. Our findings reveal that GPT-4V and LLaVA-NeXT achieve only marginal improvements over a random predictor (20%), obtaining accuracies of 25.7% and 22.3%, respectively. Gemini 1.5 Pro, designed specifically for long-context multimodal understanding, obtains an accuracy of 37.3%, which, while better, is still substantially lower than the average performance of human experts at 85.0%. These results suggest that while the multimodal community has made meaningful progress, a significant gap remains to be bridged before these systems can match human-level long-form video understanding capabilities. Progress in long-form video understanding could enable new applications including AR assistants, embodied agents, and interactive video platforms. We hope that HourVideo will serve as a benchmark to facilitate research in this direction and enable the development of multimodal models that can understand endless streams of visual data.

## 2 Benchmark Design and Construction

While open-ended question answering closely emulates human interaction, automating the evaluation of free-form natural language responses remains challenging. Given that our primary goal is to assess long-form video-language understanding capabilities, we opt for a five-way multiple-choice question-answering (MCQ) task. This approach simplifies the evaluation process by allowing to calculate an aggregate question-answering accuracy metric. In the following section, we describe our task suite and question-answer generation pipeline in detail, both of which are designed to curate diverse high-quality five-way multiple-choice questions (MCQs).

### 2.1 Task Suite

Creating a comprehensive benchmark for long-form video-language understanding is challenging, primarily because formulating meaningful questions that require processing and synthesizing information across various temporal segments is highly nontrivial, even for expert human annotators. Moreover, we note that even benchmarks for image or short video clip understanding are difficult to construct. As a result, we typically observe two common strategies for benchmark creation: (1) pre-defined label spaces testing for a specific skill or within narrow domains (e.g., Kinetics [9] and Something-Something [15]); or (2) gluing together different datasets, each designed to test a specific model capability [16–19]. In contrast, a single benchmark that can comprehensively test a suite of model capabilities can significantly benefit the research community.

We draw inspiration from both lines of research methodologies and introduce a novel suite of tasks designed to benchmark long-form video-language understanding capabilities for one-hour-long videos. Our task suite encompasses a comprehensive set of perceptual and cognitive tasks, including summarization, perception (recall, tracking), visual reasoning (spatial, temporal, predictive, causal, counterfactual), and navigation (room-to-room, object retrieval) tasks. Our strategy draws inspiration from the two common approaches previously discussed: (1) designing narrowly focused question prototypes to significantly streamline the question-answer creation process, and (2) creating a diverse suite of tasks that holistically evaluate a broad spectrum of multimodal capabilities. Our task suite with manually designed question prototypes are shown in Table 1. In particular, there are 18 sub-tasks in our proposed task suite and example MCQs from HourVideo are shown in Fig. 1.

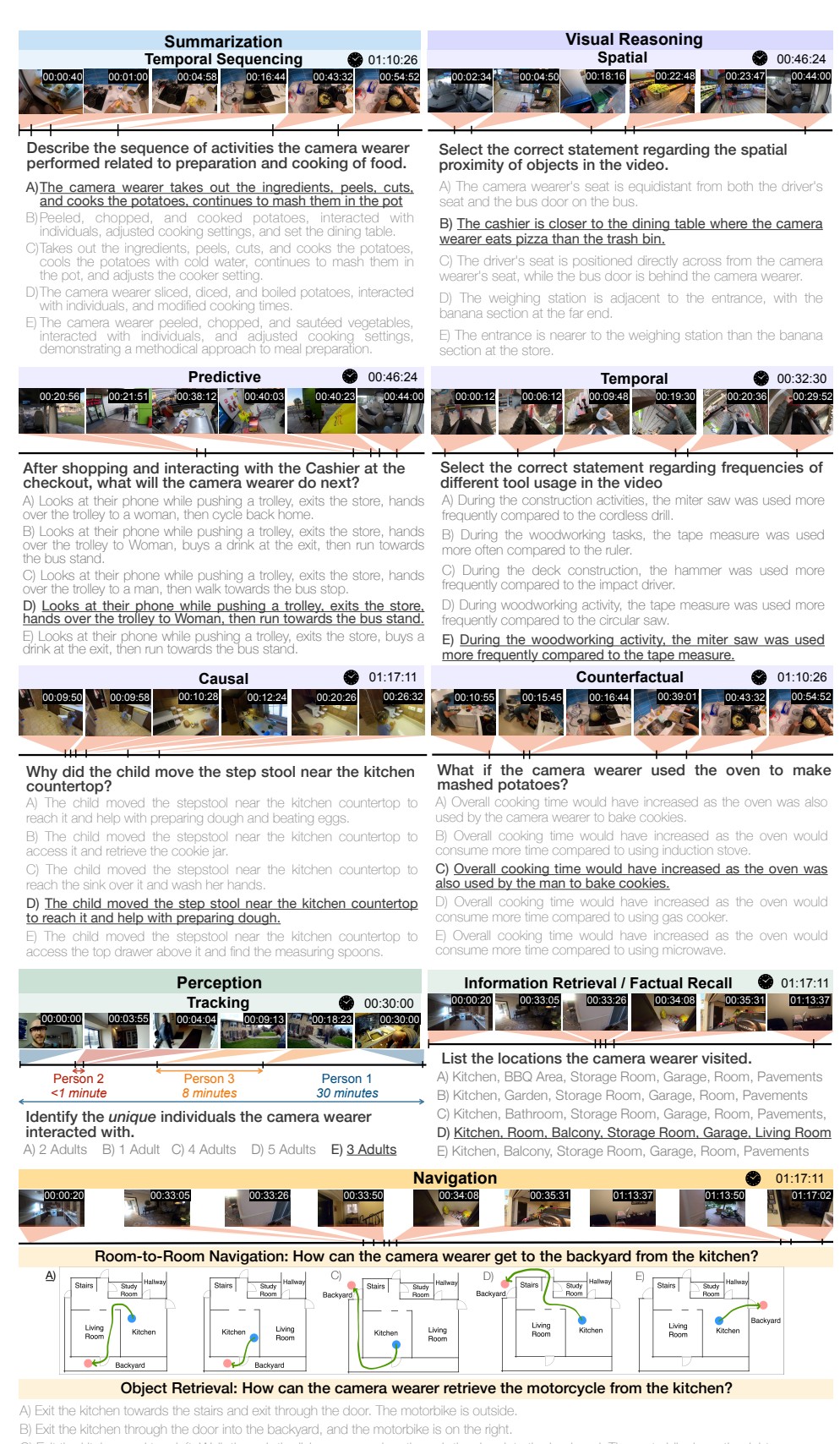

Figure 1: **Example MCQs from HourVideo** for different tasks. The correct answers are underlined.

| Summarization | |
|---|---|
| **Key Events/ Objects** | *Summarize the key interactions of the camera wearer in the [supermarket].* |
| **Temporal Sequencing** | *Describe the sequence of activities performed by the camera wearer to [prepare the dessert].* |
| **Compare/ Contrast** | *How did the camera wearer's activities in the [apartment] differ from those in the [restaurant]?* |
| **Perception** | |
| **Information Retrieval** | |
| ● Factual Recall | *What [dairy products] did the camera wearer [pick up] in the [supermarket]?* |
| ● Sequence Recall | *What did the camera wearer do immediately after [weighing tomatoes] at the [supermarket]?* |
| ● Temporal Distance | *How long after starting to [eat pizza] did the camera wearer [dispose of the pizza box]?* |
| **Tracking** | *List the unique [individuals] the camera wearer interacted with at the [drugstore].* |
| **Visual Reasoning** | |
| **Spatial** | |
| ● Relationship | *Where was the [microwave] placed in relation to the [stove] in the [kitchen]?* |
| ● Proximity | *Is the [microwave] closer to the [fridge] compared to the [sink]?* |
| ● Layout | *Which is the correct [IMAGE] depicting the layout of the camera wearer's [apartment]?* |
| **Temporal** | |
| ● Duration | *Which activity did the camera wearer spend more time on: [cooking] or [playing the piano]?* |
| ● Frequency | *Did the camera wearer use the [circular saw] or [crosscut saw] more frequently to [cut wood]?* |
| ● Pre-requisites | *What preparation steps did the camera wearer take before [baking cookies]?* |
| **Predictive** | *What is the most likely activity the camera wearer will do next after [doing laundry]?* |
| **Causal** | *Why did the camera wearer [leave the garage for the second time]?* |
| **Counterfactual** | *What if the camera wearer used the [oven] to [cook mashed potatoes]?* |
| **Navigation** | |
| **Room-to-Room** | *How did the camera wearer get from the [building entrance] to the [apartment]?* |
| **Object Retrieval** | *How can the camera wearer retrieve the [TV remote] if they are in the [kitchen]?* |

Table 1: **Our proposed task suite with question prototypes.** This table shows all 4 tasks and 18 sub-tasks proposed in **HourVideo**, along with the corresponding handcrafted question prototypes designed to evaluate long-form video-language understanding capabilities.

## 2.2 Dataset Generation Pipeline

In this section, we provide an overview of the question-answer creation pipeline that we developed to create HourVideo. The pipeline is summarized in Fig. 2.

**Video curation, Stage 1.** A crucial design consideration for this benchmark is the selection of video sources and types. We chose the Ego4D [13] dataset for our videos for multiple reasons: (1) its egocentric perspective aligns well with the typical visual input for autonomous agents and assistants; (2) it features extensive visual narrations, which aid in creating diverse multiple-choice questions; and (3) it is readily accessible under the Ego4D license. We manually reviewed 1,470 videos, ranging from 20 to 120 minutes, from the Ego4D dataset, assessing their potential to generate relevant questions for various tasks in our task suite. We engaged five human experts for video curation. Following this process, we curated 500 egocentric videos.

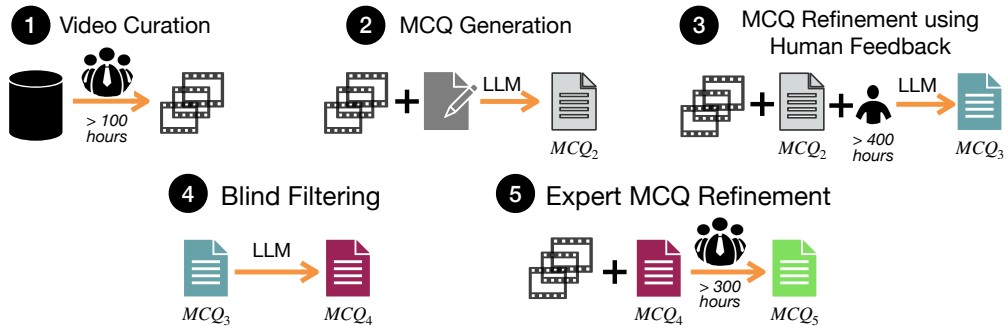

Figure 2: **Our dataset generation pipeline.** We develop a dataset generation pipeline consisting of five stages to create HourVideo. We leverage over *800 hours of human effort* in total corresponding to Video curation (Stage 1), MCQ Refinement using Human Feedback (Stage 3) and Expert MCQ Refinement (Stage 5) stages. We use LLMs for MCQ Generation (Stage 2), MCQ Refinement using Human Feedback (Stage 3) and Blind Filtering (Stage 4). We note that causal, counterfactual and navigation questions are manually generated by human experts (See Sec. 2.2 for details).

**Candidate MCQ Generation, Stage 2.** The objective of this stage is to produce high-quality MCQs for each task, requiring analysis and synthesis of information across multiple temporal segments in a long-form video. Initially, we manually develop question template(s) for each task in the suite. As shown in Table 1, transforming a question template into an actual question involves incorporating video-specific information tailored to the task and template. To facilitate this, we utilize the detailed narrations from the Ego4D dataset, transforming them into a structured format that can be processed by an LLM. Specifically, we segment the video at 20-minute intervals, with each segment's representation including a summary and a list of tools, food items, technology, humans, pets, and physical locations encountered by the camera wearer in the video. We note that synthesizing a structured representation and a question template into a valid question with correct and incorrect answers presents a significant challenge, even for advanced LLMs. Consequently, for each task, we formulate detailed prompts that offer question prototypes, comprehensive instructions, in-context examples, and step-by-step guidance on how to transform a question template into a valid candidate $MCQ_2$. In total, we developed 25 task-specific prompts.

MCQ **Refinement with LLMs using Human Feedback, Stage 3.** The purpose of this phase is to refine $MCQ_2$, created in the previous stage. $MCQ_2$ may contain invalid questions, incorrect answers, trivial incorrect options, and various other issues. We identified that a significant source of these issues stemmed from relying on the noisy narrations in Ego4D. For example, different narrators within the same video could refer to a dishwasher as a "plate rack" or use other terms, and an individual might be described as an "adult," "person with a red and white shirt," "man Y," or "teenager" at various times in the narration. These inconsistencies, combined with our automatic question generation in the first stage, could lead to generation of invalid MCQs. To address noisy MCQs, we implement a human feedback system where trained annotators are tasked with: 1) assessing the validity of each question to ensure it aligns with the video content, 2) verifying the accuracy of the given answer—if found incorrect, they provide the correct answer in free-form text, 3) ensuring that all incorrect options are factually wrong and clearly distinguishable from the correct answer. We gather human feedback for all $MCQ_2$, involving over 400 hours of human effort. We then design prompts, to automatically refine $MCQ_2$ using this human feedback to produce $MCQ_3$. We engaged seven trained annotators in this stage.

**Blind filtering, Stage 4.** Modern LLMs possess extensive prior knowledge and can thus easily answer certain questions without needing to analyze the videos. The objective of this phase is to eliminate questions that can be answered through prior knowledge or can be trivially answered without requiring any information from the video. To address this, we do blind filtering of $MCQ_3$, utilizing two separate blind LLMs (GPT-4-turbo and GPT-4). Specifically, we exclude any MCQ that is correctly answered by at least one LLM without video input. Although this method may aggressively remove MCQs, it ensures that the remaining $MCQ_4$ are of high quality and specifically tailored to test long-form video-language understanding.

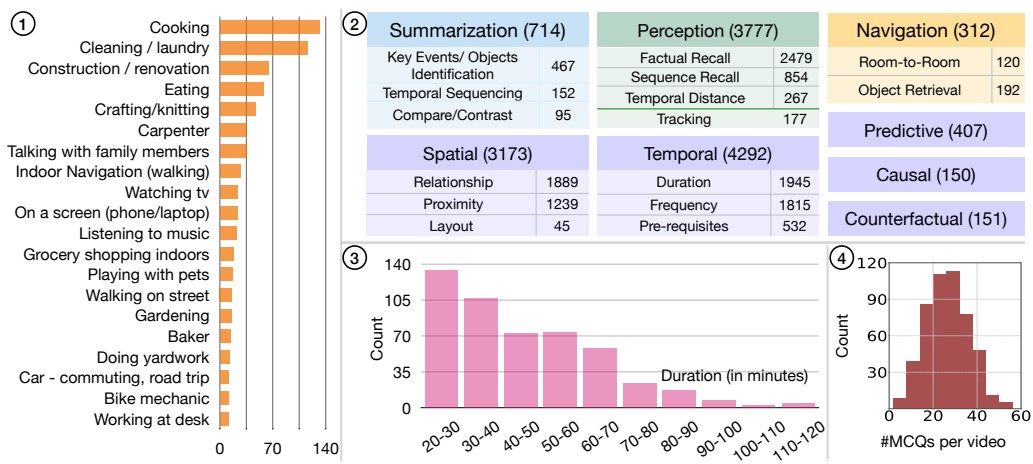

Figure 3: **Dataset Statistics.** ①: HourVideo includes 500 videos sourced from the Ego4D dataset, spanning 77 everyday scenarios. The bar chart shows the top 20 scenarios. ②: We report the number of MCQs per task/sub-task. In total, there are 12,976 questions in HourVideo. ③: We show the distribution of video duration in HourVideo. The average duration of videos in HourVideo is 45.7 minutes, with 113 videos extending beyond one hour. ④: We show the distribution of number of MCQs per video. On average, each video contains 26 MCQs.

**Expert Refinement, Stage 5.** The aim of this stage is to enhance the quality of $MCQ_4$ by utilizing a selected group of expert human annotators. This stage serves as a comprehensive step to address various remaining issues that might have persisted through prior stages. Examples of expert refinement include transforming a broad question like "Where did the camera wearer leave the keys?" into a more precise query: "Where did the camera wearer leave the bike keys after returning home from shopping?" Over 300 hours of expert human effort are employed in this stage to carefully examine and refine $MCQ_4$, culminating in a high-quality $MCQ_5$. We engaged four human experts in this stage.

**Manual Generation.** Despite our extensive efforts to automate fully or partially, we discovered that certain tasks did not align well with the pipeline we described earlier. Specifically, for causal, counterfactual, spatial layout and navigation tasks, we found it more effective to manually generate questions with human experts rather than processing through our multi-stage pipeline. Consequently, for these tasks in our benchmark, we generated high-quality questions, albeit in a smaller quantity. Four human experts were engaged in this stage, generating a total of 658 MCQs (5.1%).

**Implementation details.** We used GPT-4 in our pipeline as it offers impressive capabilities to follow complex multi-step instructions. We used the Chain-of-Thought [20] prompting strategy and a temperature of 0.1 for all stages involving LLMs in our pipeline. We show an example MCQ life-cycle in Fig. B.2. See Supplementary B for more details on dataset generation. We include the exact prompts used for generating $MCQ_2$ for the following tasks:● Narration compilation (Fig. E.1), ● Summarization (Fig. E.2, E.3), ● Perception/ Information Retrieval/ Factual Recall (Fig. E.4, E.5), ● Visual Reasoning/ Spatial/ Relationship (Fig. E.6, E.7).

## 2.3 HourVideo Statistics

HourVideo consists of 500 videos from the Ego4D dataset, covering 77 *daily life scenarios* such as cooking, cleaning, eating, watching TV, baking, etc. (Fig. 3). The dataset includes 381 hours of video footage, with video durations ranging from 20 to 120 minutes (Figure 3). On average, each video is approximately 45.7 minutes long, which $15\times$ larger than prior work in long-form video-language understanding [12]. Additionally, 113 videos in our dataset exceed one hour in length. Each video is accompanied by an average of 26 high-quality, five-way multiple-choice questions, totaling 12,976 questions in the dataset. Finally, we strive to ensure an even distribution of MCQs across all tasks in our suite, with the exception of causal, counterfactual, and navigation tasks, where questions were manually generated for a selected group of videos.

# 3 Experiments

## 3.1 Evaluation Protocol

HourVideo includes five-way multiple-choice questions, for which we report accuracies per task and in aggregate across the entire dataset. A significant challenge in evaluating MCQs over long videos is preventing information leakage across questions. Ideally, each MCQ should be evaluated independently to avoid this issue, but unfortunately, this approach is computationally expensive and time-consuming. Therefore, for our evaluation, we assess the questions in batches, with each batch containing all questions related to a specific task or sub-task. For predictive tasks (reasoning), we provide precise timestamps to trim the videos for targeted evaluation. Details on tasks and sub-tasks requiring independent evaluation are provided in Supplementary B.

## 3.2 Baselines

In this section, we compare the performance of different multimodal models on understanding long videos in a zero-shot setting. Specifically, we evaluate three classes of models: (1) Blind LLMs, (2) Socratic Models [21], and (3) Native multimodal models. All these models operate under a common function $A = M(V, \tau, Q)$ where $V, \tau, Q, M, A$ refer to the long-form video input, prompt (instruction), multiple-choice question, multimodal model, and text output respectively.

**Blind LLMs.** Modern LLMs possess extensive prior knowledge, enabling them to easily answer certain questions without the need to analyze videos. Furthermore, it is likely that some questions can be trivially answered by exploiting biases in the question-answer pairs. The 'blind' LLM baseline is designed to evaluate this by asking the LLM to answer the multiple-choice question without considering any visual information from the video, i.e., $A = M(\tau, Q)$, where $\tau$ is a generic task-agnostic prompt prepended to the question $Q$. We use GPT-4 [22] as our LLM for this baseline.

**Socratic Models.** Most current state-of-the-art multimodal models are unable to process very long videos. Therefore, to benchmark these models, we use the Socratic models approach [21]. In this approach, the video $V$, with a total duration of $t$ minutes, is segmented into one-minute intervals, each denoted as $V[i]$ for minute $i$. Each segment $V[i]$ is independently captioned, yielding a sequence of captions $z_1, z_2, z_3, \ldots, z_t$, where $z_i = \text{Video-Captioner}(V[i])$. These captions are aggregated to form a comprehensive language-based representation of the video, referred to as the world state history, which includes timestamps. This textual representation, along with a generic task-agnostic prompt $\tau$, serves as the input for long-form video-question answering: $A = M([\tau, z_1, z_2, \ldots, z_t, Q])$. We sample one-minute video clips at a rate of 0.5 fps and a resolution of $512 \times 384$. We test using both GPT-4 [22] and LLaVA-NeXT-34B-DPO [14] as the Video-Captioner. Finally, we use GPT-4 for actual question answering, as LLaVA-NeXT-34B-DPO does not support the extended context length required to process our world state history.

**Native Multimodal Models.** Multimodal video models, such as Gemini 1.5 Pro [3], are trained *jointly* on multimodal data, including audio, video, images, and text. These models are particularly adept at handling very long context lengths (2M+), making them ideal for end-to-end evaluation using our benchmark. Evaluating these models is straightforward, as they can directly process hour-long videos as $A = M(V, \tau, Q)$. For all experiments, we use a sampling rate of 0.5 frames per second, a resolution of $512 \times 384$, and a temperature setting of 0.1.

**Human performance.** Due to the high costs associated with human evaluations, we sampled 14 videos from our benchmark, which included more than 18 scenarios in total including crafting/painting, cooking, construction/renovation, gardening, cleaning/laundry and yard work. We ask three human experts to conduct evaluations on 11.2 hours of video content, encompassing a total of 213 MCQs. To prevent any contamination, we ensured that human experts who evaluated videos were not involved in the annotation of the same videos at any earlier stage (Stages 3 and 5 discussed in Sec. 2). The human experts achieve an accuracy of **85.0%**. The results are shown in Fig. 4.

## 3.3 Results

We report all task and sub-task level quantitative results in Tab. 2. Qualitative evaluations, including human evaluation numbers, are presented in Fig. 4. We remark that random guessing corresponds to 20% accuracy. Below, we discuss our key observations.

| | Summarization | | | Perception | | | | Visual Reasoning | | | | | | | | | Navigation | | Avg. |
|---|---|---|---|---|---|---|---|---|---|---|---|---|---|---|---|---|---|---|---|
| | Key Events/ Objects | Temporal Sequencing | Compare/ Contrast | Factual Recall | Sequence Recall | Temporal Distance | Tracking | Relationship | Proximity | Layout | Duration | Frequency | Pre-requisites | Predictive | Causal | Counterfactual | Room-to-Room | Object Retrieval | |
| **Blind LLMs** | | | | | | | | | | | | | | | | | | | |
| GPT-4 | 22.7 | 29.6 | 24.2 | 21.9 | 15.2 | 20.6 | 15.8 | 14.9 | 21.4 | 22.2 | 23.6 | 19.3 | 14.7 | 14.5 | 18.7 | 21.2 | 15.8 | 18.8 | 19.6 |
| **Socratic Models** | | | | | | | | | | | | | | | | | | | |
| LLaVA-34B-DPO | 34.0 | 35.5 | 35.8 | 30.3 | 19.3 | 12.7 | 34.5 | 18.3 | 15.3 | 26.7 | 21.3 | 17.9 | 23.5 | 20.9 | 21.3 | 22.4 | **20.8** | 22.4 | 22.3 |
| GPT-4 | 40.5 | 41.5 | 43.2 | 33.1 | 20.0 | **20.2** | **36.7** | 18.5 | 21.7 | **37.8** | 25.3 | 22.9 | 27.1 | 24.1 | 24.7 | 26.5 | 20.0 | 26.6 | 25.7 |
| **Multimodal Models** | | | | | | | | | | | | | | | | | | | |
| Gemini 1.5 Pro[*] | **56.4** | **59.5** | **46.7** | **41.8** | **33.6** | 19.7 | 35.7 | **27.4** | **38.2** | 21.4 | **37.2** | **35.4** | **46.8** | **46.3** | **41.0** | **38.7** | 19.2 | **33.9** | **37.3** |

Table 2: **Baseline results on HourVideo.** We report results for Blind LLMs (GPT-4), Socratic models with GPT-4 and LLaVA-NeXT-34B-DPO video captions, and Gemini 1.5 Pro. Gemini 1.5 Pro outperforms Blind LLMs and Socratic LLMs by a significant margin across all tasks (14 out of 18 sub-tasks). We remark that all these approaches rely on generative models. Prompts used for evaluation are available at hourvideo.stanford.edu.

**Blind LLMs vs. Socratic LLMs.** On aggregate, blind LLMs achieve an accuracy of 19.6%, indicating that our benchmark requires access to video content for effective performance. Comparing Blind LLMs and Socratic models, both variants of Socratic models perform marginally better than blind LLMs. It is worth noting that the GPT-4-based Socratic model approach performs considerably better on the summarization task (41.1%) than blind LLMs (24.4%) and LLaVA-NeXT-34B-DPO (34.6%). Qualitative comparisons are shown in Fig. 4.

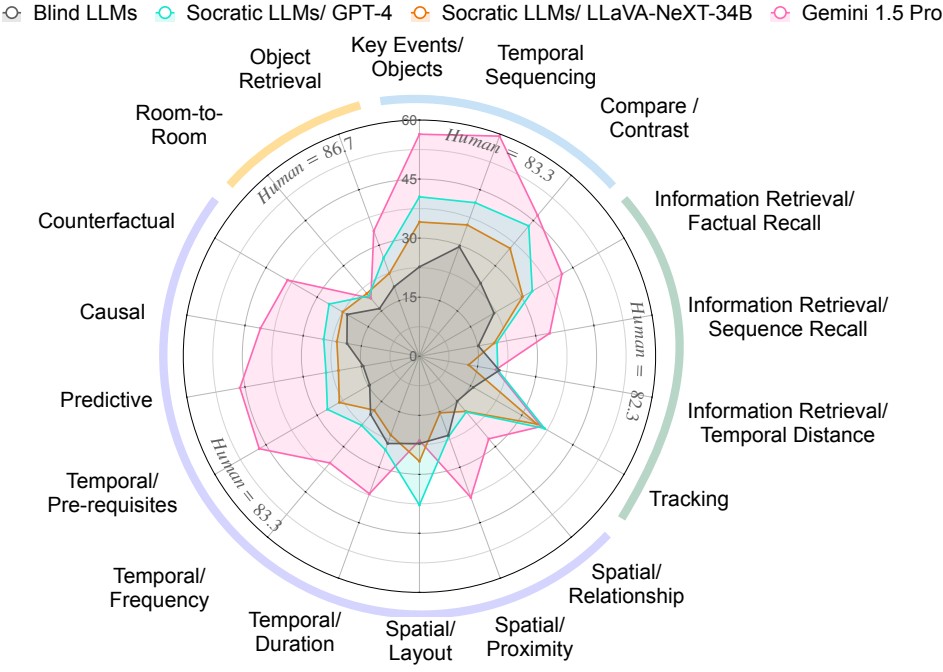

Figure 4: Comparison between different multimodal foundation models on HourVideo across different tasks/sub-tasks. We include human expert performance for summarization (83.3%), perception (82.3%), visual reasoning (83.3%) and navigation (86.7%) tasks. As one can observe, current multimodal models significantly lack long-form video-language understanding capabilities.

[*]Gemini 1.5 Pro evaluation includes 445 videos, covering 10,842 MCQ. For details, see Supplementary D.2.

**Socratic models vs. Native Multimodal Models.** Gemini 1.5 Pro outperforms Socratic models by a considerable margin across all 4 tasks–summarization, perception, visual reasoning, and navigation–indicating that similar models may be promising avenues toward long-form video-language understanding. On aggregate, Gemini 1.5 Pro outperforms the GPT-4-based Socratic model by 11.6%. Despite these significant improvements, it is important to note that Gemini's performance, at 37.3%, still lags significantly behind that of human experts, who achieve 85.0%.

**Independent vs. Task-level `MCQ` evaluation.** To investigate the validity of our proposed task/sub-task level evaluation method, we conducted an ablation study where each multiple-choice question (MCQ) was evaluated independently. For this, we used 15.9 hours of video and 570 MCQs across 25 randomly selected videos. We used Gemini 1.5 Pro, which demonstrated the highest performance on HourVideo (37.3%). The results and evaluation costs are shown in Tab. 3. There is a minor drop (2.1%) in performance when evaluating each MCQ independently; however, the associated costs increase by more than threefold. These results highlight the efficiency and validity of our proposed task-level/subtask level evaluation method.

We will require benchmark submissions to indicate whether they used task-level or individual `MCQ` evaluation when submitting their results, allowing for greater transparency and comparability between methods.

|  | Performance | Total Tokens | Evaluation Cost |
|---|---|---|---|
| Task-level | 38.9% | 120,818,343 | $846 |
| Individual | 36.8% | 374,396,885 | $2621 |

Table 3: Performance and evaluation cost comparison for our proposed task/sub-task level vs. individual `MCQ` evaluation.

# 4 Related Work

**Dataset Comparison.** Existing video benchmarks [23, 24, 11, 25–32], primarily focus on specific domains or short videos, which limit their ability to assess long-form video understanding comprehensively. Efforts like WebVid10M [33], InternVid [34], and Panda-70M [35] include detailed captions to provide video pretraining data but consist primarily of short video clips less than one minute in length and do not provide QA pairs. Recent works have introduced several benchmarks specifically designed for long video understanding, such as Next-QA [36], Next-GQA [37], VideoChatGPT [38], EgoSchema [12], MovieChat-1K [39] and MovieNet-QA [40]. [41] introduced benchmarks for evaluating relational space-time query tasks. Perception Test [42] proposed a diagnostic benchmark for multimodal models, probing for memory, abstraction, physics, and semantic capabilities using short video clips (23s average duration). However, the average video length in these datasets is still relatively short, with Ego-Schema having an average duration of 3 minutes. In contrast, we focus on hour-long video-language understanding, with videos averaging 45.7 minutes in duration (Table 4) and tasks requiring long-term comprehension.

**Video Understanding Tasks.** Significant efforts have been made to design tasks appropriate for evaluating multimodal large language models (MLLMs) [43–51]. The evaluation of vision-language models (VLMs) focuses mainly on visual perception tasks such as image-text matching, retrieval, captioning, object detection, and visual grounding tasks) [45–47]. Methods revolving around contrastive learning

| Benchmark | # Videos | Avg. len. (mins) | # Questions |
|---|---|---|---|
| MSRVTT-QA [23] | 2,990 | 0.25 | 72,821 |
| ActivityNet-QA [11] | 800 | 1.85 | 8,000 |
| TVQA [25] | 2,179 | 0.19 | 15,253 |
| How2QA [26] | 1,166 | 0.25 | 2,852 |
| NExT-QA [36] | 1,000 | 0.66 | 8,564 |
| EgoSchema [12] | 5,063 | 3.0 | 5,063 |
| **HourVideo** | **500** | **45.7** | **12,976** |

Table 4: **Dataset statistics** comparison between video understanding benchmarks.

on image-text pairs have proven to be effective methods for learning transferable representations for these visual tasks [52–54], and have been shown to be effective in more specific domains such as multi-disciplinary scientific understanding [50, 55] and multi-modal mathematical reasoning [48, 49]. Later work has improved upon the visual reasoning capabilities of VLMs [1, 56–63] and their ability to reason across complex spatio-temporal video data [64–71]. To better evaluate spatio-temporal abilities, specific benchmarks [12, 28, 31, 72, 73] have been developed. However, the questions in

many of these datasets are often not challenging enough to fully evaluate the capabilities of models in understanding long-form video content and can often be answered from only a single frame [74]. In contrast, our benchmark focuses on evaluating the capabilities needed to reason over a significantly longer duration and with more sophisticated reasoning. The questions in our dataset are designed to be highly challenging, with novel video question categories such as navigation highlighting our benchmark's ability to effectively assess the limitations of current state-of-the-art multimodal models in comprehending long-form videos.

**Long-Form Video Understanding.** To extend video-language models [75–84] to long-form videos, the main challenge lies in efficiently encoding the temporal and spatial dynamics over a long horizon. One widely used strategy is to maintain a memory bank to store history information in long videos [85–91]. Alternatively, other methods have been proposed to compact spatio-temporal tokens into a smaller set of compressed or merged tokens to reduce redundancy and alleviate computational burden [39, 81, 92–98]. Another line of work leverages language as a bridge by first generating textual descriptions for shorter video clips sub-sampled from the longer video and then employing an LLM to aggregate the short captions for longer video understanding [99–101]. In contrast, approaches like TimeChat [102] and VTimeLLM [103] aim to enhance temporal localization capabilities by encoding timestamp knowledge into visual tokens or using multi-stage training methods. Despite these extensive efforts, long-form video understanding remains a significant challenge for the current generation of multimodal models.

## 5 Conclusion

We introduce **HourVideo**, a novel benchmark dataset designed to rigorously evaluate the capabilities of multimodal models to comprehend one-hour-long videos. Our dataset consists of a novel task suite comprising summarization, perception (*recall, tracking*), visual reasoning (*spatial, temporal, predictive, causal, counterfactual*), and navigation (*room-to-room, object retrieval*) tasks. This benchmark includes 500 egocentric videos from the Ego4D dataset, spanning durations of 20 to 120 minutes, and features 12,976 high-quality five-way multiple-choice questions. Our zero-shot evaluation on HourVideo reveal that multimodal models such as GPT-4V and LLaVA-NeXT exhibit performance levels only slightly better than random guessing. In stark contrast, human expert performance substantially surpasses state-of-the-art long-context multimodal model Gemini 1.5 Pro (85.0% accuracy versus 37.3%), highlighting significant research gap. We aim to establish HourVideo as a benchmark challenge to spur the development of advanced multimodal models capable of truly understanding endless streams of visual data.

**Limitations and future work.** Despite our substantial efforts to create a high-quality benchmark dataset, there may still be some inconsistencies within the multiple-choice questions. Additionally, while this is currently the largest long-form video-language understanding benchmark of its kind to the best of our knowledge, we acknowledge the need for more holistic benchmarks that include diverse video sources such as sports and YouTube videos. Lastly, we note that incorporating support for the audio modality is essential for more comprehensive evaluation of multimodal models. We also remark that our world extends beyond visual and auditory stimuli to include other sensory modalities (e.g., tactile), suggesting opportunities to explore these additional modalities in future work. We discuss broader impact of HourVideo in Supplementary E.

## Acknowledgments and Disclosure of Funding

This work was in part supported by the Stanford Institute for Human-Centered Artificial Intelligence (HAI), ONR N00014-23-1-2355, and Microsoft. This work was supported by API credit grants from Google DeepMind and OpenAI. We thank Vishal Dharmadhikari for assistance with setting up Gemini 1.5 evaluations, Hashem Elezabi and Canon Grace Pham for help with data curation. We thank Chengshu (Eric) Li and Sanjana Srivastava for discussions on navigation questions, and Michael Poli, Daniel Y Fu, Jing Yu Koh, Stephen Tian, Tristan Thrush and Ngoc-Trung Tran for their feedback on the manuscript. We also thank our reviewers for their comments.

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

## HourVideo Supplementary Material

## A   HourVideo Release v1.0

We are releasing HourVideo v1.0, our proposed benchmark dataset for one-hour video-language understanding. The benchmark dataset is provided as a single JSON file for ease of use and for straightforward integration with existing benchmarking pipelines. For each video, the dataset includes metadata and contains multiple-choice questions covering multiple tasks from our proposed task suite. Each task is accompanied by a set of multiple-choice questions, each with five possible answers. For predictive visual reasoning tasks, relevant timestamps are provided to allow precise video trimming. Additionally, a PyTorch dataloader is provided to efficiently load the video and the benchmark dataset. We provide all the 500 video_uids used in our benchmark, and users can simply download the corresponding videos from the Ego4D website after reviewing and accepting the Ego4D license agreement. We provide 2 sample videos with annotations from HourVideo. All materials are available at hourvideo.stanford.edu.

- **Structure**: HourVideo v1.0 release is organized as follows :
  - **data/**
    * `HourVideo_v1_0.json`: Contains all 12976 questions in the benchmark dataset.
    * `navigation_task_images/`: Contains all images which are part of the navigation task.
    * `spatial_layout_task_images/`: Contains all images which are part of the spatial layout (reasoning/spatial) task.
    * `sample_annotations/`: Given that **HourVideo** is an evaluation benchmark, ground truth annotations will not be released to public. For review purposes, we provide ground truth annotations for 2 sample videos as csv files.
    * `csv/`: We provide the benchmark in individual csv files for each video to enhance accessibility, allowing users to conveniently view the contents for each video separately.
  - **src/**
    * `video_utils.py`: A script for video processing functionalities.
    * `hourvideo_dataloader.py`: A PyTorch DataLoader script designed to efficiently load and preprocess the dataset.
    * `baselines/`: Contains all prompts and code for captioning/ question answering for Blind LLMs, Socratic models and Multimodal Video Models. **Remark:** Except for LLaVA-NeXT-34B-DPO captioning experiments, all other experiments require access to proprietary models including GPT-4 and Gemini 1.5 Pro.
- **Documentation**: We provide a comprehensive datasheet explaining the benchmark dataset's purpose and intended usage.
- **License**: HourVideo will be made publicly available under Apache 2.0 License. Do note that Ego4D videos are publicly available under the Ego4D License [13].
- **Versioning and Updates**: We will maintain HourVideo, with all updates and new versions announced publicly.
- **Contact Information**: For additional inquiries, please contact `keshik@stanford.edu`.

# B  Data Generation Pipeline: Additional details

## B.1  Prompt Design

We meticulously designed 25 prompts in total for tasks/ sub-tasks in our proposed task suite. For 9 out of 15 tasks, we generate questions first, followed by jointly generating answers and wrong answers. For the predictive visual reasoning and temporal pre-requisites tasks, we jointly generate questions and answers first, followed by generating wrong answers. For causal, counterfactual, spatial layout and navigation tasks, we generate questions, answers and wrong answers manually. We also designed prompts for narration compilation (See Fig. E.1) and paraphrasing answers for the summarization, temporal pre-requisites, and predictive visual reasoning tasks. We show the exact prompts used for the following tasks for question-answer generation (Stage 2):● Narration compilation (Fig. E.1), ● Summarization (Fig. E.2, E.3), ● Perception/information_retrieval/factual_recall (Fig. E.4, E.5), and ● Visual_reasoning/spatial/proximity (Fig. E.6, E.7). For more details, refer to hourvideo.stanford.edu.

## B.2  Narration Compilation Details

We segment all our videos at 20 minute intervals and extract a semi-structured representation which includes `title`, `description`, `start_identifier`, `end_identifier`, `list of tools`, `list of food items`, `list of technology objects`, `list of humans interacted`, `list of pets interacted` and `list of unique locations` in the video segment.

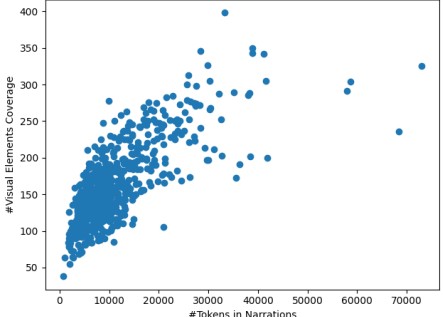

Finally, these segments are compiled by a LLM to form a single structured representation for each video. The prompt is shown in Fig. E.1. Considering that Ego4D offers two independently collected sets of narrations for each video, we select the narration set with the higher token count. This design choice is based on our empirical observation that a larger number of tokens typically ensures more comprehensive coverage of visual elements. These results are shown in Fig. B.1.

## B.3  Human Feedback and Expert Refinement Details

For *MCQ Refinement with Large Language Models using Human Feedback (Stage 3)*, we engaged seven annotators who had been trained to provide human feedback based on examples created by our team. Continuous quality assessments were conducted throughout this stage to ensure the integrity and high quality of the feedback obtained for `MCQ` Refinement. More than 400 hours of human effort were spent in this stage. For

Figure B.1:  This plot shows visual elements coverage vs. total number of narration tokens. We use collection of objects in ImageNet-21K, VisualGenome, Tencent1M and Places365 to quantify visual coverage. We use Tiktoken library to calculate the total number of tokens. We used Ego4D dataset [13] to perform this experiment.

*Expert Refinement (Stage 5)*, we engaged four human experts dedicating over 250 hours of human effort for *QAW Refinement*.

# C  Evaluation details

**Evaluation Protocol.** As discussed in Sect. 3, we have developed an evaluation protocol that assesses multimodal models at the level of individual tasks and sub-tasks. The specific tasks and sub-tasks requiring independent evaluation are listed in Tab. C.1.

This structured approach minimizes information leakage across questions and mitigates the substantial costs associated with individual `MCQ` evaluation. It is important to note that the costs of individual `MCQ` evaluation are proportional to the number of questions, emphasizing the need for our proposed assessment strategy.

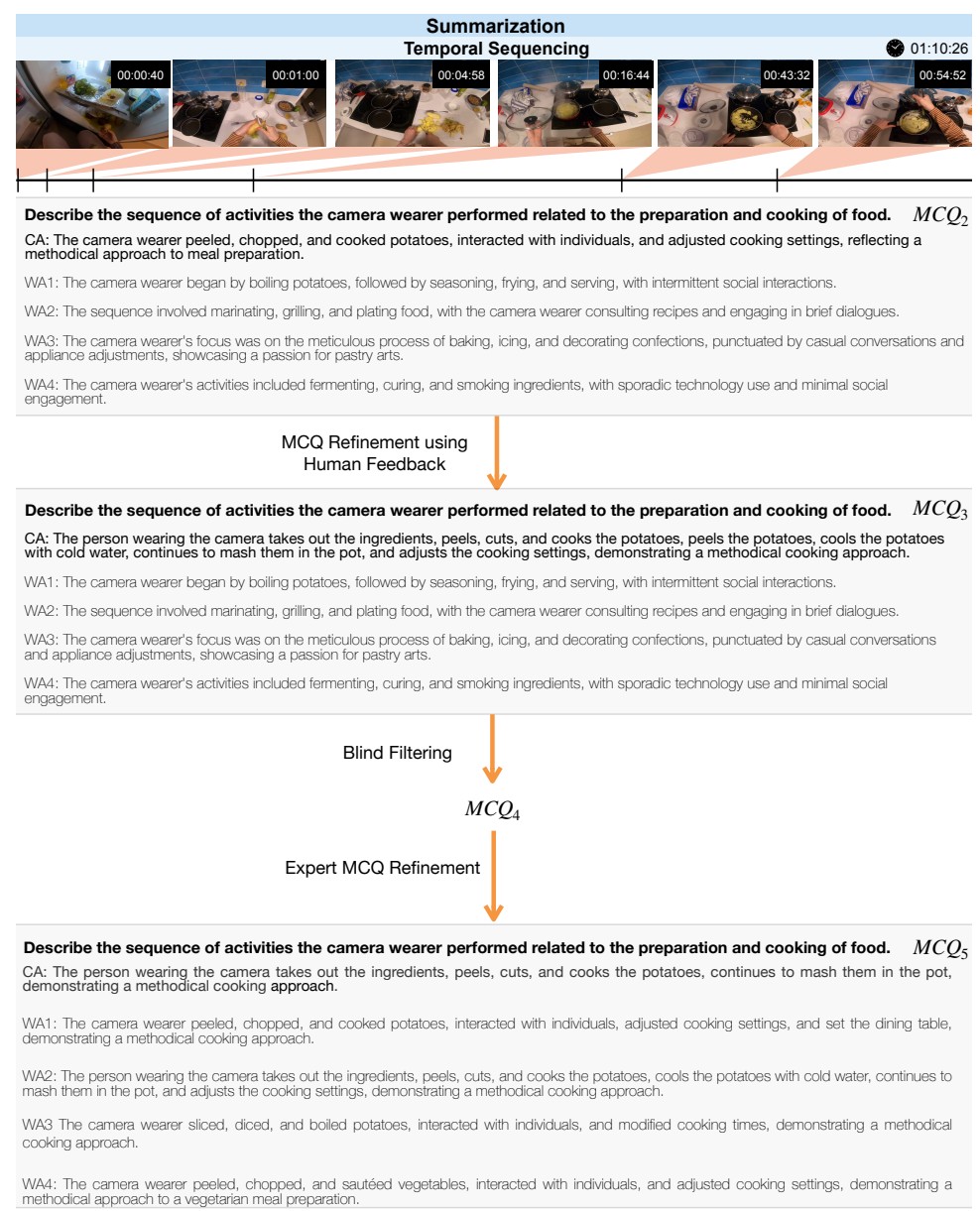

Figure B.2: MCQ **life-cycle example.** We show an example from our dataset generation pipeline, depicting the transformation of MCQs through successive stages, resulting in MCQ$_5$. Notably, the Expert Refinement Stage is allowed to perform rigorous modifications on the MCQs, including filtering, grounding questions and answers, and refining both correct and incorrect answers to increase the complexity and challenge of the final MCQs.

# D   Additional Experiments

## D.1   Additional Baselines

We conduct an additional experiment using the recently released Tarsier model [104] which reports state-of-the-art results in multiple short-form video understanding benchmarks. Following the exact setup in Tarsier for long-video understanding, we use the publicly available Tarsier-7B model with 16 frames uniformly sampled from the entire video. The results are reported in Tab. D.1. We remark that Ego4D [13] is used in Tarsier pre-training (Video captioning task). **Prompts.** For all our baseline experiments, we use a generic task-agnostic prompt together with the video and MCQ tests for evaluation. All our prompts for baseline evaluations are included in the evaluation toolkit. We leave advanced prompting strategies for future work.

| Parent Task | Sub-task (if any) | Node (if any) |
|---|---|---|
| Summarization | | |
| Perception | Information Retrieval | Factual Recall |
| | | Sequence Recall |
| | | Temporal Distance |
| Perception | Tracking | |
| Visual Reasoning | Spatial | Relationship |
| | | Proximity |
| | | Layout |
| | Temporal | Duration |
| | | Frequency |
| | | Pre-requisites |
| | Predictive | |
| | Causal | |
| | Counterfactual | |
| Navigation | Room-to-Room | |
| | Object Retrieval | |
| | Room-to-Room (Image-based) | |
| | Object Retrieval (Image-based) | |

Table C.1: **The table shows our proposed evaluation protocol**. Tasks and sub-tasks requiring independent evaluation are  highlighted.

| | Summarization | Perception | Visual Reasoning | Navigation | Avg. |
|---|---|---|---|---|---|
| Blind LLMs | | | | | |
| GPT-4 | 24.4 | 20.0 | 19.1 | 17.6 | 19.6 |
| Socratic Models | | | | | |
| LLaVA-NeXT-34B | 34.6 | 26.7 | 19.1 | 21.8 | 22.3 |
| GPT-4 | 41.0 | 29.4 | 22.8 | 24.0 | 25.7 |
| Multimodal Models | | | | | |
| Gemini 1.5 Pro | 55.8 | 38.2 | 35.7 | 28.1 | 37.3 |
| SOTA short-form video model | | | | | |
| Tarsier-7B (16 frames) | 32.2 | 24.7 | 27.4 | 17.9 | 26.7 |

Table D.1: **Additonal results on HourVideo using Tarsier-7B [104].** Tarsier-7B (16 frames) performance is comparable to Socratic LLMs. We remark that the Ego4D [13] dataset is used in the pre-training stage of the Tarsier model for video captioning.

## D.2 Model Refusal Rates

Proprietary models, such as GPT-4 and Gemini 1.5 Pro can abstain from responding to MCQs for various reasons, including video content filtering, privacy concerns, and other undisclosed factors. In particular, we observed that the model refusal rates were significantly higher for Gemini 1.5 Pro compared to GPT-4. For Socratic models, both GPT-4 and LLaVA-34B-DPO models successfully caption more than 96% of the 1-min segments. We report refusal rates for question-answering in Tab. D.2.

| Model | Videos/MCQs answered | Refusal rate |
|---|---|---|
| GPT-4 (Blind) | 500 / 12,930 | 0.35% |
| GPT-4 (Socratic) | 500 / 12,959 | 0.13% |
| LLaVA-34B-DPO (Socratic) | 500 / 12,953 | 0.18% |
| Gemini 1.5 Pro | 445 / 10,842 | 16.45% |

Table D.2: Model refusal rates: We report refusal rates for various models for 500 videos / 12,976 MCQs. For Socratic LLMs, we report the refusal rates for question answering. The refusal rate for Gemini 1.5 Pro is significantly higher compared to GPT-4.

# E Broader Impact

The Long-form Video-Language Understanding Benchmark (HourVideo) introduced in this work has the potential to significantly advance the field of AI video understanding and enable a wide range of useful applications. By focusing on long-form video, HourVideo challenges models to demonstrate high-level reasoning and comprehension skills that more closely mirror human intelligence. Success on this benchmark could lead to AI systems that can effectively perceive and interact with the real world over extended periods of time, unlocking transformative capabilities in areas like embodied AI and robotics, autonomous vehicles, smart environments, and augmented/virtual reality.

Embodied AI and robotics, which aim to develop artificial agents that can perceive, navigate, and physically interact with their environment, could benefit greatly from advances in long-form video understanding. A robot or embodied agent that can maintain a coherent, long-term understanding of its surroundings and goals would be far more capable and adaptable than one operating with only short-term perception. It could handle more complex, multi-stage tasks, learn from extended observations, and build rich mental models to support planning and decision making. For example, a home robot with long-form video understanding could tidy up a room by keeping track of object locations, understanding the steps involved in cleaning tasks, and adapting to unexpected obstacles or messes. Similarly, an industrial robot with long-term video comprehension could perform intricate assembly tasks, monitor and maintain complex machinery, or collaborate seamlessly with human workers. Long-form video understanding is thus a key missing piece in realizing the full potential of embodied AI and robotics.

Progress on HourVideo could also contribute to the development of large world models – AI systems that learn comprehensive, multi-modal representations of the world from vast amounts of data. By processing and consolidating information from extended video sequences, these models could construct more complete and coherent world knowledge that spans time and integrates multiple levels of abstraction. Long-form video understanding would allow these models to not just recognize isolated snapshots, but grasp the flow of events, the persistence and transformation of objects, the rules of physics and causality, and the complex interactions between agents and their environments. This deep, temporally-informed world knowledge could in turn support more advanced reasoning, prediction, planning, and generalization.

Long-form video understanding is also crucial for creating compelling augmented reality (AR) and virtual reality (VR) experiences. An AR system that can parse and adapt to a user's visual context over time would be a far more capable assistant than one that merely labels objects frame-by-frame. In VR, AI characters and environments that evolve responsively to a user's choices and actions throughout an extended interactive session would provide a deeper sense of immersion and realism.

While these exciting applications underscore the importance of advancing long-form video understanding, it is equally critical to consider the potential risks and ethical implications involved. Video data, particularly long-running egocentric video as used in HourVideo, can be highly sensitive and revealing of personal details. As AI video understanding capabilities grow, robust safeguards must be put in place to protect individual privacy, ensure secure data handling, maintain transparency around data collection and use, and prevent unauthorized surveillance or abuse. The intimate window that AR/VR systems and embodied AI agents have into users' private spaces and behaviors further heightens these concerns. As world models become more comprehensive and powerful, it will be crucial to ensure they are developed and used in ways that respect privacy, promote fairness and transparency, and align with human values.

In designing HourVideo, we have taken care to use only videos that are licensed for research and to focus the benchmark on high-level semantic understanding rather than invasive personal information extraction. Nonetheless, the overarching trajectory toward machines that can deeply interpret the visual world will require ongoing vigilance and proactive efforts to align their development and deployment with societal values.

In summary, HourVideo offers a valuable step forward for AI video understanding, with promising implications for embodied AI, robotics, large world models, AR/VR, and beyond. However, for long-form video understanding technology to realize its full positive potential, the AI research community must prioritize the responsible development of these powerful capabilities with strong commitments to ethics, privacy, security, and beneficial impact for humanity. We believe our benchmark will shape

the progress of video understanding systems to be not only more capable, but also more trustworthy and socially beneficial.

**Potential Negative Societal Impact.** Advancements in **HourVideo** benchmark could significantly enhance AI capabilities towards building autonomous agents. However, these technologies could also, for example, fuel the development of more sophisticated surveillance systems, raising significant privacy concerns. While such advancements have potential security benefits, they pose risks if used inappropriately, threatening individual privacy in public and private spaces. It is crucial that developments in video understanding are accompanied by stringent ethical standards and robust privacy safeguards to prevent misuse.

**Amount of Compute.** We report the total amount of compute used for captioning 381 hours of video content using LLaVA-NeXT-34B-DPO in Table E.3. For GPT-4, we spent a total of ≈$10,000 in credits which includes the entire dataset generation pipeline and baseline experiments (Blind LLMs and Socratic LLMs). Gemini 1.5 Pro baseline experiments cost approximately $105 per one-hour video across all tasks/sub-tasks.

Table E.3: Amount of compute/ API usage used in this project. The GPU hours include computations for initial explorations/ prompt engineering / experiments to produce the reported values. CO2 emission values are computed using https://mlco2.github.io/impact/

| Experiment | Hardware | GPU hours | Carbon emitted in kg |
|---|---|---|---|
| Main paper : Table 2 (LLaVA-NeXT-34B) | A6000 | 120 | 9.00 |
| Main paper : Table 2 (LLaVA-NeXT-34B) | RTX A5000 | 24 | 1.66 |
| Additional Compute for Hyper-parameter tuning | RTX A5000 | 12 | 1.80 |
| **Total** | | **156** | **12.46** |

## Narration Compilation

**MAIN INSTRUCTIONS:**
In the "Long-Form Egocentric Video Narrative Compilation" task, you are working with detailed narrations
from long-form, real-world, egocentric videos.
Your goal is to compile these narrations into a chronological narrative that accurately reflects the linear progression of events in
the video, from the perspective of 'C', the camera wearer ...............

For each segment, structure your summary as follows:
```json
  {
    "segment_title": "<Generated Title>",
    "segment_description": "<Generated summary>",
    "segment_start_identifier": "<Starting Unique Identifier>",
    "segment_end_identifier": "<End Unique Identifier>",
    "segment_tool_list": "<List of tools used in the segment>",
    "segment_food_list": "<List of food related items used in the segment>",
    "segment_technology_list": "<List of technology related objects used in the segment>",
    "segment_humans_list": "<List of humans/pets that C interacted with in the segment>",
    "segment_pets_list": "<List of pets that C interacted with in the segment>",
    "segment_locations_list": "<List of specific, named locations that C visited or mentioned in the segment>"
  }
```

**CHRONOLOGICAL NARRATIVE COMPILATION GUIDELINES:**
  1. Concise Segment-Based Narrative Construction:
  ...........................
  2. Clarity, Brevity, and Object Emphasis in Language:
  ...........................
  3. Narrative Integrity and Object Relevance:
  ...........................
  4. Objective and Efficient Representation:
  ...........................

**STRICTLY AVOID:**
...........................

**ADDITIONAL CONSIDERATIONS:**
.......................

**FORMATTING INSTRUCTIONS:**
...........................

**EXAMPLE OUTPUT:**
[
  {
    "segment_title": "Initial Activities in Living Room and Kitchen",
    "segment_description": "'C' starts in the living room and then moves to the kitchen.
    She stands up, walks around, interacts with a man named K, and uses her phone. In the kitchen, 'C' opens the fridge,
    takes out potatoes, and then moves to the kitchen counter where she begins to peel potatoes.,
    highlighting the use of technology and tools like a phone and knife, and the involvement of Man K and a dog.",
    "segment_start_identifier": "T0000_0",
    "segment_end_identifier": "T0010_17",
    "segment_tool_list": ["Phone", "Knife", "Fridge"],
    "segment_food_list": ["Potatoes"],
    "segment_technology_list": ["Phone"],
    "segment_humans_list": ["Man K"],
    "segment_pets_list": ["Dog"],
    "segment_locations_list": ["Kitchen", "Living Room"]
  },
  .............................
]

**DENSE NARRATIONS:**
<Insert Video Narrations below>

Figure E.1: **Our Narration Compilation Prompt** We show the prompt used for Narration Compilation task. This prompt is designed to compile dense narrations to a structured format, providing step-by-step instructions, formatting guidelines and output examples for narration compilation. The dense narrations are obtained from Ego4D [13].

**Summarization: Question Generation**

**MAIN INSTRUCTIONS:**
In the "In-depth Analysis of Extended Videos" class, students will analyze video action narrations divided into chronological segments.Your task is to develop two to five challenging questions about the high-level details in the video and requires watching the entire video to answer. These questions should encourage students to comprehend and think critically about the video's narrative and key elements. Note that in the video narrations provided, 'C' refers to the camera wearer ...........

Each segment is structured as follows:
```json
  {
    "segment_description": "<Summary>",
    .....................
  }
```

**SKILL-FOCUSED INSTRUCTIONS:**
  Your questions should test students' ability to:
  - Summarize/ comprehend the video content entirely.
  - Compare and contrast different sections of the video to understand its development.
  - Distill the video's content to identify central concepts or themes.

**QUESTION GENERATION GUIDELINES:**

**1.** Strive to Use Each Question Type below where I have provided examples for each type: Aim to generate one question from each of the following type. You may skip a question type only if it is genuinely incompatible with the video content:
**- Key Events Identification:** "Summarize the key interactions and events in the video, focusing on their impact and significance."
**- Key Tools Identification:** "What tools were used by the camera wearer when repairing the bike?"
**- Key Technology Use Identification:** "What actions did the camera wearer take that involved the use of technology, and how did these actions fit into the overall sequence of events?"
**- Temporal Sequencing:** "Describe the sequence of activities the camera wearer performed in the kitchen related to food preparation."
**- Compare and Contrast:** "Compare and contrast the activities the camera wearer engaged in within the apartment and outside of it."
Alternatively, feel free to create similar questions based on the narration content.
**........**
**8.** Please use 'the camera wearer' instead of 'C' to emphasize the egocentric perspective in all questions.

**STRICTLY AVOID THE FOLLOWING TYPES OF QUESTIONS:**
**1.** "When ...?"
2. "How many ...?"
3. "How much ...?"
.........
**8.** PLEASE avoid any references to the time of day when generating the questions (night-time, morning time, bed tme etc.).

**FORMATTING INSTRUCTIONS:**
STRICTLY Return your output as a list of dictionaries as shown in the EXAMPLE OUTPUT below.

**EXAMPLE OUTPUT:**
[
  {
    "question": "Summarize the key interactions and events in the video, focusing on their impact and significance.",
    "question_type": "Key Events Identification"
  },
  ………………..
  ]

**CHRONOLOGICAL NARRATION SEGMENTS:**
<Insert Narration Compilation below>

Figure E.2: **Summarization question generation prompt.** We show the prompt used for generating summarization questions. This prompt includes question prototypes, step-by-step instructions, formatting guidelines and in-context examples for generating summarization questions. The narration compilation is obtained using the prompt in Fig. E.1.

**Summarization: Answer/ Wrong Answers Generation**

**Instructions for creating challenging MCQ Test for students based on provided questions.**

**Main Instructions:**
I want you to act as a teacher in the class called "Long-term video understanding." I will provide video narrations along with their timestamps, and a set of highly difficult questions for your students about the high-level details in the video. In this task, I want you to create a difficult Multiple Choice Question (MCQ) exam that tests the following abilities of students:
- Ability 1: Students' ability to summarize and compare long parts of the video.
- Ability 2: Students' ability to compress information from the video rather than just listing the actions that happened in the video, and to analyze the subtleties and complexities within the video.
- Ability 3: Students' ability to identify the most important parts of the video, and how these important parts are interconnected and evolve throughout the video.

Note: Please note, in the video narrations provided, 'C' represents the camera wearer, capturing their perspective and actions.

**Objective:**
Your objective is to create a challenging MCQ exam testing the above abilities, based on the provided questions. Given the extended length of the video, ensure that the answers require a comprehensive understanding of the entire video content. Please STRICTLY follow the steps below.

**Steps:**
Step 1: Review the Provided Questions:
……..

Step 2: Create the Correct Answer for Each Question using Video Narrations:
……

Step 3: Develop Four Misleading Wrong Answers for Each Question:
……

Step 4: Ensure Relevance and Avoid Direct References:
……

Step 5: Format the Answers:
……

Step 6: Repeat Steps 1 to 5 for all the remaining questions in the provided list:
……

**Step-by-Step Example for your Reference:**
……

**Guidelines**
1. Only Output Step 6 Results.
2. STRICTLY Maintain consistency in answer length: All answers, including the correct one, should be of similar length to ensure fairness.
3. Strictly avoid terms including "such as, etc." in your answers. Craft all answers very objectively, leaving no space for interpretations.
4. AVOID giving away hints that identify incorrect answers.
5. STRICTLY stay faithful to narrations.
6. Please use 'the camera wearer' instead of 'C' to emphasize the egocentric perspective in all questions.
7. NEVER modify any of the questions.

**List of Questions:**
<GENERATED QUESTIONS>

**CHRONOLOGICAL NARRATION SEGMENTS**
<NARRATION COMPILATION>

Figure E.3: **Summarization answer/ wrong answers generation prompt.** In this Figure, we show the prompt used for generating answers and wrong answers for summarization questions. This prompt includes instructions, step-by-step examples, formatting guidelines for generating answers/ wrong answers for our summarization questions. The questions are generated using the prompt shown in Fig. E.2. The narration compilation is obtained using the prompt in Fig. E.1.

**Perception / Information Retrieval / Factual Recall: Question/ Answer  Generation**

Instructions for Generating Factual Recall Questions and Answers for a College-level Course:

**Objective:**
You are tasked with creating one to eight highly challenging questions that test memory recall of specific details from an egocentric video. You are challenged to create 3 types of questions:
- **Type I:** Object Attributes: Questions that test on Object Attributes such as color, shape etc. i.e., "What color sweater was the camera wearer wearing?"
- **Type II:** Counting: Questions that test Counting. i.e., "How many potatoes did the camera wearer take from the refrigerator?"
- **Type III:** Object-Action: Questions that test for specific object(s) used in a unique and singular action. Provide answers where possible, or mark as <Cannot Determine> if the answer is not explicit in the narration.

Note that 'C' refers to the camera wearer in the narration.

Each segment is structured as follows:
```json
{
    "segment_description": "<Summary>",
    ................................
}
```

**Steps:**
- Step 1: Analyse Segments and Identify Unique Objects/Actions:
...............................

- Step 2: Question Formulation Based on Object Details and Actions:
Objective: Create questions that:
For Type I, delve into the attributes of objects (color, shape etc).
For Type II, involve counting items (how many of a certain object are there).
For Type III, focus on identifying specific objects used in notable UNIQUE/SINGULAR actions.
...............................

- Step 3: Question and Answer Compilation with Reference Identifiers:
...............................

- Step 4: Verify that both the Question and Answers are interesting and non-trivial.
...............................

- Step 5: Iteration for Additional Questions:
...............................

**Step-by-Step Example for Reference:**
...............................

**GENERAL GUIDELINES FOR THIS TASK.**
1. Only output Step 5 Results.
...............................
10. Strive to generate Questions of all three types mentioned in the instructions.

**STRICTLY AVOID THE FOLLOWING TYPES OF QUESTIONS:**
...............................

**The Video Narration is below:**
<Video Narrations>

Figure E.4: **Factual Recall (Perception/Information Retrieval) question/ answer generation prompt.** We show the prompt used for generating factual recall questions. This prompt includes question prototypes, step-by-step instructions, formatting guidelines and in-context examples for generating factual recall questions. There are three types of questions generated (Object attributes, Counting and Object-Action relationship). The dense narrations are obtained from Ego4D [13].

**Perception / Information Retrieval / Factual Recall: Wrong Answers Generation**

**Instructions for Generating Factual Recall Questions and Wrong Answers for a College-level Course:**

**Objective:**
Your task is to create multiple choice questions (MCQs) for a college-level course, focusing on recalling specific details from a video narration. The twist in this task is to generate four plausible wrong answers for each question, testing students' detailed knowledge and understanding of the video content while fostering critical thinking.

Note that 'C' refers to the camera wearer in the narration.

**Steps:**
**- Step 1: Review Question Details:**
  Carefully analyze each generated question, focusing on object attributes, counting, and object-action.
  Understand the factual background to effectively create misleading yet plausible wrong answers.

**- Step 2: Protocol for Correct Answer:**
  Each MCQ will provide the correct answer annotated by humans (labelled as "answer").
  STRICTLY Use the provided "answer" as the "Correct Answer" without modification.

**- Step 3: Develop Four Plausible Wrong Answers:**
  For each question, create four wrong answers that are:
  (i) Plausible within the context of the video but intentionally incorrect.
  (ii) Varied, addressing different aspects or common misconceptions related to the question's focus.

**- Step 4: Ensure and Craft Closely Related Wrong Answers:**
  Craft four wrong answers for each question that are plausible yet intentionally incorrect,
  without indirectly hinting at the correct answer. Provide a variety of plausible scenarios to challenge
  the students to critically evaluate their knowledge against the video content.

**- Step 5: Review the Correct Answer Protocol and Wrong Answers:**
  Please review if the generation process has satisfied Steps 1 - 4. If not, re-generate again.

**- Step 6: Format Questions and Wrong Answers:**
  Organize each question with its four wrong answers in a clear and concise structured format,
  ensuring the answers challenge the student's understanding without being directly correct.
  Please see the example below for reference.

**- Step 7: Iteration for Additional Questions:**
  Repeat the process for all remaining QA pairs.

**STRICTLY FOLLOW THIS EXPECTED OUTPUT FOR YOUR REFERENCE FROM STEP 7:**
…………………………..

**GENERAL GUIDELINES FOR THIS TASK.**
1. Output Step 7 Results.
2. All wrong answers must be plausible to someone who has not viewed the video, avoiding overly far-fetched options.
3. Challenge students to critically evaluate their recall of the video by providing closely related but incorrect alternatives.
4. Maintain clarity and simplicity in crafting answers to avoid ambiguity.
5. Use generic descriptions and common knowledge to construct answers, steering clear of specifics that could inadvertently reveal the correct answer.
6. Diversify the wrong answers to cover a broad spectrum of plausible inaccuracies related to the question's topic.
7. NEVER modify any of the Question.

**List of Questions:**
<List of Generated Questions with Answers>

**CHRONOLOGICAL NARRATION SEGMENTS**
<VIDEO NARRATIONS>

Figure E.5: **Factual Recall (Perception/Information Retrieval) wrong answers generation prompt.** We show the prompt used for generating wrong answers for factual recall questions. This prompt includes instructions, step-by-step examples and formatting guidelines for generating wrong answers. The questions/ answers are generated using the prompt shown in Fig. E.4. The narration compilation is obtained using the prompt in Fig. E.1.

## Visual Reasoning / Spatial / Relationship: Question Generation

**Instructions for Generating Spatial Relationship Questions for Static Objects with Unique Identifiers:**

**Introduction:**
These narrations come from a long-form, real-world, egocentric video, showcasing the linear progression of events from the perspective of 'C', the camera wearer. The narrations provide contextual clues essential for understanding the spatial layout of specific environments. Each narration is accompanied by unique identifiers.

Each segment is structured as follows:
```json
{
  "segment_description": "<Summary>",
  …………………………..
}
```

**Objective:**
Generate two to five interesting spatial relationship questions based on static objects within a single environment using these narrations. Ensure the selected objects are mentioned at least twice in the narration to confirm their significance. Note the unique identifiers associated with environments and objects in the narration.

**Steps**:
**- Step 1: Infer a Single Environment with Identifier:**
Description: Analyze the narration to infer a single environment such as a kitchen or living room. Note the unique identifier for this environment (e.g., T0203_3 for the kitchen). If the narration does not clearly describe a single environment, stop and output "[{"question": "Cannot Determine Environment"}]".

**- Step 2: Infer Static Objects in the Environment with Identifiers:**
Description: Identify static objects present in the inferred environment and note their unique identifiers (e.g., microwave T0204_4, fridge T0205_5 in the kitchen). If the objects are moving in the video, STRICTLY avoid choosing them. Ensure each object is mentioned at least twice in the narration to confirm its significance. If suitable objects aren't mentioned at least twice, stop and output "[{"question": "Cannot Determine Static Objects"}]".

**- Step 3: Select Two "Interesting/ Significant" Static Objects:**
Description: From the list of identified objects, select two that are mentioned at least twice and are particularly interesting or significant, and note their unique identifiers.

**- Step 4: Formulate Spatial Relationship Question with Identifiers:**
Description: Pose a question about the spatial relationship between the two selected objects using their unique identifiers. Example: "Where is the microwave (T0204_4) located in relation to the fridge (T0205_5)?" If this cannot be formulated based on the narration, stop and output "[{"question": "Cannot Generate Questions"}]".

**- Step 5: Explore More Static Objects in the same environment / Explore More Static Environments:**
Description: If the narration provides clear descriptions of other static objects/ environments, repeat the steps for those settings to generate additional questions.

**Step-by-Step Example for Reference:**
…………………………..

**Guidelines:**
1. Only output Step 5 Results.
…………………………..
6. Please cater your questions in simple English so that non-native English speakers can understand.

**Video Narration:**
<Narration Compilation>

Figure E.6: **Spatial relationship (visual reasoning) question generation prompt.** We show the prompt used for generating spatial relationship questions. This prompt includes question prototypes, step-by-step instructions, formatting guidelines and in-context examples for generating spatial relationship questions. The narration compilation is obtained using the prompt in Fig. E.1.

**Visual Reasoning / Spatial / Relationship: Answer/ Wrong Answers Generation**

**Instructions for Creating Challenging MCQ Test with Intentionally Incorrect Answers:**

**Main Instructions:**
Description:
As a teacher in the class called "Long-term Video Understanding," your task is to create a unique and challenging Multiple Choice Question (MCQ) exam. You are provided with video narrations and timestamps, but will creatively devise a "Dummy" Correct Answer and several Wrong Answers for questions on spatial relationships, such as "Where is the microwave located in relation to the stove?" All answers are designed to be incorrect, yet plausible, to encourage critical evaluation.

**Objective:**
Develop an MCQ exam that tests students' understanding of spatial relationships between objects or locations in a video, with the twist that all answer options, including the one labeled as "correct," are intentionally incorrect. This fosters critical thinking and skepticism.

**Steps**:
**- Step 1: Fabricate a Plausible "Dummy" Correct Answer:**
Invent a plausible yet intentionally incorrect answer. This answer should be convincing as a possible correct answer without familiarity with the video content.

**- Step 2: Develop Four Misleading Wrong Answers:**
Create four additional incorrect but plausible answers, distinct from the "Dummy" Correct Answer, using varied spatial terms and relationships. Remember that answers should be pratical. Answers like "microwave on top of fridge", "fridge is under the cooker" are superficial, so strictly avoid such answers.

**- Step 3: Ensure Plausibility and Consistency**
All answers, while incorrect, should be plausible and consistent with common knowledge about spatial relationships in typical environments. You are welcome to use the video narrations for help. But remember all answers should NEVER BE SUPERFICIAL. Your answers SHOULD contain "left", "right" relationships wherever possible. NEVER use objects/ locations that are not mentioned in the questions when crafting answers. Iterate if required.

**- Step 4: Clearly Format Your Answers**
Present the question, the "Dummy" Correct Answer, and the four wrong answers in a clear and consistent dictionary format, as shown in the example. Please see the example below for reference.

**- Step 5: Repeat for All MCQs**
Fabricate plausible but incorrect answers for each MCQ, aiming to challenge students through strategic variation and complexity.

**- Step 6: Compile Results**
Return all fabricated MCQs in a list of dictionaries, adhering to the demonstrated format below.

**STRICTLY FOLLOW THIS EXPECTED OUTPUT FOR YOUR REFERENCE FROM STEP 6:**
…………………………..

**GUIDELINES**
1. Only Output Step 6 Results.
2. Strictly avoid terms including "such as, etc." in your answers. Craft all answers very objectively, leaving no space for interpretations.
……….
10. Please keep all your answers simple and clear.

**List of Questions:**
<List of Generated Questions>

**CHRONOLOGICAL NARRATION SEGMENTS**
<NARRATION COMPILATION>

Figure E.7: **Spatial Relationship (Visual Reasoning) answer/ wrong answers generation prompt.** We show the prompt used for generating answer/ wrong answers for spatial relationship questions. This prompt includes step-by-step instructions, formatting guidelines, generated questions and narration compilation for generating answer/ wrong answers. Questions are generated using the prompt shown in Fig. E.6, while the narration compilation is obtained using the prompt in Fig. E.1.

