# OpenReview forum: "HourVideo: 1-Hour Video-Language Understanding"
_NeurIPS.cc/2024/Datasets_and_Benchmarks_Track — NeurIPS 2024 Track Datasets and Benchmarks Poster_

### Official Review · Reviewer_Qx1u · 2024-07-21
**Review of HourVideo**

**Rating:** 5
**Confidence:** 4
**Correctness:** Yes.
**Clarity:** Yes.

**Review:**

The paper provides a high-quality and comprehensive benchmark for long-duration video understanding, featuring diverse tasks and meticulous annotations. However, it relies on multiple-choice questions, which might oversimplify evaluation, and lacks evaluations of open-sourced Video LLMs like MovieChat. Additionally, there are missed ablation studies on independent QA evaluation and end-to-end experiments. Despite these drawbacks, the paper addresses a critical gap in multimodal model capabilities and has the potential for significant real-world impact.

**Strengths:**

- Comprehensive Benchmark: The dataset covers a wide range of tasks that genuinely require long-term comprehension, making it a valuable resource for advancing multimodal model capabilities.
- High-Quality Annotations: The questions are meticulously crafted to ensure they require information synthesis across multiple temporal segments, enhancing the benchmark's robustness.
- Diverse Task Suite: The inclusion of tasks like visual reasoning and navigation tests a model’s cognitive abilities in real-world scenarios, providing a holistic evaluation.

**Additional Feedback:**

- Minor Problem:
  - In Line 47, `12,500` should be `13,000`.
  - It will be better to cite Perception Test which proposes similar evaluation dimensions (`Perception Test: A Diagnostic Benchmark for Multimodal Video Models`).

**Documentation:**

Yes.

**Ethics:**

No.

**Limitations:**

Yes.

**Opportunities For Improvement:**

- Performance Metrics: The reliance on multiple-choice questions might oversimplify the evaluation, not fully capturing the nuanced understanding required for long-term video comprehension.
- Limited Evaluations: More open-sourced Video LLMs should be evaluated for the benchmark, especially those can input long videos with memory, e.g., MovieChat.
- Missed Ablation Studies:
  - In Line 162, the authors claim that the QAs are not evaluated individually. Are there any differences if QAs are evaluated independently?
  - In this paper, the authors do not directly evaluate the Video LLMs in an end-to-end way. What about the results of inputing only 32 or 64 frames sparsely sampled from the whole videos?

**Relation To Prior Work:**

Yes.

**Summary And Contributions:**

The paper presents HourVideo, a benchmark dataset designed for long-duration video-language understanding tasks. The dataset comprises 500 egocentric videos from the Ego4D dataset, spanning 20 to 120 minutes, and features 13,000 high-quality multiple-choice questions. HourVideo evaluates models on tasks such as summarization, perception, visual reasoning, and navigation. Initial results show that state-of-the-art multimodal models perform poorly compared to human baselines, highlighting a significant research gap in long-term video understanding.

---

> ### Author Rebuttal · Authors · 2024-08-17
>
> ## Response to Reviewer Qx1u (Part 2/2)
>
> (Continuation)
>
> > Missed Ablation Studies: Are there any differences if QAs are evaluated independently?
>
> We appreciate your insightful comment. As outlined in our evaluation protocol (Main - Sec 3.1, Supplementary - Sec. C.1), evaluating each question individually offers more granularity but comes with significantly higher computational costs. To address reviewer's concern, we conducted an **ablation study covering 15.9 hours /570 MCQ tests across 25 randomly selected videos**, where each multiple-choice question (MCQ) was evaluated independently. We chose to use the Gemini 1.5 Pro model, which demonstrated the highest performance on HourVideo (39.9%). The results, including the evaluation costs are presented below:
>
> |                           | Performance | Total Tokens | Evaluation Cost |
> | ------------------------- | ----------- | ------------ | --------------- |
> | Task-level MCQ Evaluation | 38.9%       | 120,818,343  | $846            |
> | Individual MCQ Evaluation | 36.8%       | 374,396,885  | $2621           |
>
> **As these results indicate, while there is a minor drop (2.1%) in performance when evaluating each MCQ independently, the associated costs increase by more than threefold. These results emphasize the efficiency and validity of our proposed task-level/subtask level evaluation method.** Additionally, we will require participants to indicate whether they used task-level or individual MCQ evaluation when submitting their results, ensuring transparency and comparability across methods.
>
>
>
> $~$
>
> > In this paper, the authors do not directly evaluate the Video LLMs in an end-to-end way. What about the results of inputing only 32 or 64 frames sparsely sampled from the whole videos?
>
> Thank you for the comment. We would like to clarify that the **Gemini 1.5 Pro baseline model (main paper: Table 2) is evaluated in an end-to-end manner.** Gemini 1.5 Pro processes the entire video along with MCQ questions as input and directly outputs answers. For reproducibility, we have included the inference code with prompts for all our reported baselines in the Supplementary materials.
>
> We address reviewer’s request to benchmark Video LLMs with frames sparsely sampled from the entire video by conducting experiments using **Tarsier** which reports state-of-the-art results in multiple short-form video understanding benchmarks. Following the **exact** setup in Tarsier for long-video understanding, we use the publicly available **Tarsier-7B** model with 16 frames sparsely sampled from the entire video. The results for HourVideo benchmark (500 videos) are as follows:
>
> |                                  | **Summarization** | **Perception** | **Visual Reasoning** | **Navigation** | **Average** |
> | -------------------------------- | ----------------- | -------------- | -------------------- | -------------- | ----------- |
> | Blind LLMs                       | 24.3              | 20.0           | 19.2                 | 20.2           | 19.7        |
> | Socratic LLM: GPT-4             | 41.0              | 29.4           | 23.3                 | 10.9           | 25.8        |
> | Socratic LLM: LLaVA-NeXT-34B-DPO | 34.5              | 26.8           | 19.2                 | 21.8           | 22.3        |
> | Tarsier-7B (16 frames)           | 32.2              | 24.7           | 27.6                 | 17.9           | 26.9        |
> | **Gemini 1.5 Pro**               | **56.7**          | **40.8**       | **37.8**             | **35.7**       | **39.9**    |
>
> Tarsier-7B performs similar to Socratic LLMs. These additional results will be included in the final version.
>
>
> $~$
>
> > In Line 47, 12,500 should be 13,000.
>
> > It will be better to cite Perception Test which proposes similar evaluation dimensions (Perception Test: A Diagnostic Benchmark for Multimodal Video Models).
>
> Thank you. We will fix the typo and cite Perception Test.

---

> ### Author Rebuttal · Authors · 2024-08-17
>
> ## Response to Reviewer Qx1u (Part 1/2)
>
> We sincerely thank the reviewer for the comprehensive and encouraging feedback. Below, we address each concern raised by the reviewer.
>
>
> > Performance Metrics: The reliance on multiple-choice questions might oversimplify the evaluation, not fully capturing the nuanced understanding required for long-term video comprehension.
>
> In video question answering (QA), the two prevalent approaches are multiple-choice question answering (MCQ) and open-ended question answering. While open-ended QA closely emulates human interaction, automating open-ended QA evaluation is challenging. In contrast, MCQs provide a straightforward and objective evaluation metric and are widely adopted across multiple video understanding benchmarks including NextQA, How2QA, TVQA, STAR and EgoSchema.
>
> A key challenge is designing the question, answer and negative answer to ensure that high performance in benchmark requires a nuanced understanding of the events happening in the video. Towards this end, we invest **significant human effort (650+ hours)** in both Stage 3 (MCQ Refinement using Human Feedback) and Stage 5 (Expert MCQ Refinement) to generate **hard negatives** for each MCQ. Consider an example from Figure 1 below:
>
> | Describe the sequence of activities the camera wearer performed related to preparation and cooking of food (Summarization/ Temporal Sequencing) |                                                              |
> | ------------------------------------------------------------ | ------------------------------------------------------------ |
> | Multiple-choice answer                                       | Explanation                                                  |
> | A. The camera wearer takes out the ingredients, peels, cuts, and cooks the potatoes, continues to mash them in the pot. | **Correct Answer**                                               |
> | B. Peeled, chopped, and cooked potatoes, interacted with individuals, adjusted cooking settings, and set the dining table. | Incorrect: The camera wearer did not set the dining table.   |
> | C. Takes out the ingredients, peels, cuts, and cooks the potatoes, cools the potatoes with cold water, continues to mash them in the pot, and adjusts the cooker setting. | Incorrect: The camera wearer did not cool potatoes with cold water. |
> | D. The camera wearer sliced, diced, and boiled potatoes, interacted with individuals, and modified cooking times. | Incorrect: The camera wearer did not set dice the potatoes. Interacting with individuals in the video is not related to preparation/ cooking of food. |
> | E. The camera wearer peeled, chopped, and sautéed vegetables, interacted with individuals, and adjusted cooking settings, demonstrating a methodical approach to meal preparation. | Incorrect: The camera wearer did not sauté vegetables. Interacting with individuals in the video is not related to preparation/ cooking of food. |
>
> **As shown in the example above, the presence of hard negatives ensures that the model has to develop a nuanced and detailed understanding of temporal events in the video to answer the question correctly.**
>
>
>
> $~$
>
> > Limited Evaluations: More open-sourced Video LLMs should be evaluated for the benchmark, especially those can input long videos with memory, e.g., MovieChat.
>
> We agree with the reviewer that including more open-sourced LLMs would be beneficial. Given the large number of multimodal LLMs, for our baselines, we selected a representative family of methods : Blind LLMs, Socratic LLMs, and Native Multimodal LLMs. We remark that this set included the LLaVA-NeXT-34B-DPO model --used for the Socratic LLM approach-- which is open-sourced [A].
>
> Nevertheless, we address reviewer’s request by benchmarking **MovieChat** [B] on a subset of HourVideo covering 57.0 hours of duration/1533 MCQ tests/ 75 randomly selected videos. The experiment setup is as follows: LLM=Llama, fps=0.5, qa-mode=global. As shown below, MovieChat performs marginally better than random chance.
>
> |                              | Blind LLMs | Socratic LLMs/ LLaVA-NeXT-34B-DPO | Socratic LLMs/ GPT-4 | Gemini 1.5 Pro | MovieChat |
> | ---------------------------- | ---------- | --------------------------------- | --------------------- | -------------- | --------- |
> | HourVideo Evaluation Results | 19.7       | 22.3                              | 25.8                  | **39.9**       | 20.7      |
>
> Finally, our goal is to create a public leaderboard and engage actively with the community so that more methods can be benchmarked.
>
> [A] Zhang, Yuanhan, et al. *LLaVA-NeXT: A Strong Zero-Shot Video Understanding Model*. Apr. 2024
>
> [B] Song, Enxin, et al. "Moviechat: From dense token to sparse memory for long video understanding." CVPR 2024
>
> $~$
>
> (To be continued)

---

> ### Author Response · Authors · 2024-08-29
> **Seeking Feedback from Reviewer Qx1u**
>
> Dear Reviewer **Qx1u**,
>
> With the author-reviewer discussion phase concluding in just over three days, we would greatly appreciate it if you could let us know whether our responses are satisfactory.
>
> We have addressed all your concerns and conducted three additional sets of experiments as requested: Tarsier-7B, MovieChat, and an ablation study comparing task-level and individual QA evaluation. We are happy to address any further concerns you may have.
>
> We sincerely appreciate your time and effort.
>
> Thank you,
>
> Authors

---

> > ### Author Response · Authors · 2024-09-01
> > **Looking Forward to Reviewer Qx1u's Feedback**
> >
> > Dear Reviewer **Qx1u**,
> >
> > We thank the reviewer for the comprehensive feedback.
> >
> > As today is the final day of the discussion phase, we would appreciate it if you could let us know whether our responses are adequate. We’re happy to provide any further information.
> >
> > Thank you for your time.
> >
> > Best,
> >
> > Authors

---

> ### Comment · Area_Chair_Ab7z · 2024-09-01
> **Review/rebuttal discussion**
>
> Dear reviewer Qx1u,
>
> Could you please take a look at the rebuttal and let the author(s) and the other reviewers know if the rebuttal changes your opinion about the paper?
>
> Thanks

---

### Official Review · Reviewer_Z2rz · 2024-07-24
**thoughtfully constructed benchmark for evaluating one-hour video understanding**

**Rating:** 8
**Confidence:** 4
**Clarity:** yes, very detailed and clear. kudos t…

**Review:**

Quality: Comprehensive dataset presented with a detailed / thoughtful task suite, and entire pipeline to generate questions described in great detail between paper and supplemental material.

Clarity: This paper is very well organized and detailed explanations of methods / experiments are provided along the way.

Originality: This paper presents a novel benchmark over long video understanding with task diversity baked in.

Significance: The paper clearly address a gap in long for video understanding, particularly in advancing autonomous systems via ego centric videos.


Pros:

- Comprehensive and diverse dataset constructed with a well defined set of types of questions / task suite
- Addresses and demonstrates critical gap in understanding long-videos
- Benchmark has potential to spur new research directions / advances in the field

Cons:

- Dataset construction procedure is resource-intensive limiting adoption of similar mechanisms by other groups on new datasets
- Benchmark evaluations rely on existing methods/LLMs, would have been interesting to see some novel method applied to this dataset
- Restricted to Egocentric videos (though to proper audience thats probably a pro lol)

**Strengths:**

Strengths:

- Long-video understanding is a significant challenge and the benchmark demonstrates there is a significant gap with existing models compared to human performance
- Rigorous methodology and detailed explanations and evaluations
- Broad range of tasks / question types make this benchmark applicable to different real world use cases

**Additional Feedback:**

minor questions

- how do you ensure that each of the incorrect answers in the multiple choice question setting are reasonable / not obvious?
- in line 198 you mention 3 human experts conducted evaluations - what qualifies them as a human "expert"? and what does it mean that they achieve accuracy of 84.4% (is that an average? or did only one human answer each question, something else?)

**Correctness:**

Are the claims made in the submission correct? yes

If the submission is a dataset, it is constructed in a sound way? yes

If it is a benchmark, are the evaluation methods and experiment design appropriate and performed correctly? yes

**Documentation:**

For datasets, is there sufficient detail on data collection and organization, availability and maintenance, and ethical and responsible use?  yes

For benchmarks, is there sufficient detail to support reproducibility? yes

**Ethics:**

not really, mostly minor notes

authors should probably verify:

- with respect to "Fair Wages: all human research subjects or participants must receive appropriate compensation. If you make use of crowdsourcing or contract work for a particular task as part of your research project,  you must respect the minimum hourly rate in the region where the work is carried out." - since you used human labeling/experts that these people were fairly compensatesd

- with respect to "Privacy: Datasets should minimize the exposure of any personally identifiable information, unless informed consent from those individuals is provided to do so. " - that PII exposure is minimized (it appeared so in the two sample videos provided, with face blurs for example).

**Limitations:**

Have the authors adequately addressed the limitations and potential negative societal impact of their work? I believe so

Some other limitations / things to consider if you haven't already:

- The dataset is based on the Ego4D dataset, which may have inherent biases in terms of the scenarios and demographics represented. This could limit the generalizability and fairness of the benchmark in diverse real-world applications.
- The use of egocentric videos, often involving personal and private activities, raises potential privacy concerns. Ensuring that all data is anonymized and used ethically is crucial, but details on these measures are not extensively covered in the submission.

**Opportunities For Improvement:**

- i wonder if it would make sense to include benchmark numbers for whatever is SOTA in short form video understanding to demonstrate potential gaps of those models as well
- left some some nit questions in additional feedback section

**Relation To Prior Work:**

yes

**Summary And Contributions:**

This paper introduces a benchmark dataset aimed at evaluating one-hour video-language understanding. This benchmark features a task suite including summarization, perception (recall, tracking), visual reasoning (spatial, temporal, predictive, causal, counterfactual), and navigation (room-to-room, object retrieval) tasks.

Main Contributions:

- HourVideo dataset of 500 ego4D based videos and 13,000 high quality multiple choice questions
- Detailed discussion on procedure to generate the questions and motivation/rationale for suite of tasks that the questions cover, and supporting resources / prompts / etc
- Benchmark evaluation using blind LLMs, socratic method, and naive multimodal models

---

> ### Author Rebuttal · Authors · 2024-08-17
>
> ## Response to Reviewer Z2rz (Part 2/2)
>
> (Continuation)
>
> > Addressing reviewer’s comments on Biases, Privacy and Personally Identifiable Information.
>
> - **Dataset biases**: Although the curators of the Ego4D dataset have noted certain biases (e.g., biases towards urban or college town areas, at-home scenarios, and a lack of major social events), the videos in Ego4D are diverse, spanning 74 cities in 9 countries, 5 continents and 931 unique camera wearers of varying ages, nationalities, and gender identities. By leveraging Ego4D, we significantly improve our ability to accurately benchmark long form video understanding across a wide range of actors and environments.
> - **Privacy /Personally Identifiable Information:** We strongly agree with the reviewer that privacy concerns and ethical usage of data is extremely important. For all videos in our dataset, either (1) “informed consent for capturing identities is explicitly collected from all participants in the scene”, (2) the videos have been manually verified to ensure they “do not contain any personally identifiable information (PII)”, or (3) the videos have undergone extensive de-identification using “advanced video redaction software, open source tools, and hours of human reviews to redact visible PIIs”. Here, we directly quote the procedure from Appendix B of the Ego4D paper.
>
>
>
> $~$
>
> > Addressing reviewer’s comments on Fair Wages.
>
> We have included these details in the datasheet. We copy the relevant part from our datasheet for reference below:
> “The authors and contractors were involved in the data annotation process. The contractors were based in China, and were paid on average \\\$5 per hour, significantly higher than the \\\$1.27 hourly minimum wage in the country.”
>
>
>
> $~$
>
> > How do you ensure that each of the incorrect answers in the multiple choice question setting are reasonable / not obvious?
>
> We invest **significant human effort (650+ hours)** in both Stage 3 (MCQ Refinement using Human Feedback) and Stage 5 (Expert MCQ Refinement) to generate **hard negatives** for each MCQ. Consider an example from Figure 1 below:
>
> | Describe the sequence of activities the camera wearer performed related to preparation and cooking of food (Summarization/ Temporal Sequencing) |                                                              |
> | ------------------------------------------------------------ | ------------------------------------------------------------ |
> | Multiple-choice answer                                       | Explanation                                                  |
> | A. The camera wearer takes out the ingredients, peels, cuts, and cooks the potatoes, continues to mash them in the pot. | **Correct Answer**                                           |
> | B. Peeled, chopped, and cooked potatoes, interacted with individuals, adjusted cooking settings, and set the dining table. | Incorrect: The camera wearer did not set the dining table.   |
> | C. Takes out the ingredients, peels, cuts, and cooks the potatoes, cools the potatoes with cold water, continues to mash them in the pot, and adjusts the cooker setting. | Incorrect: The camera wearer did not cool potatoes with cold water. |
> | D. The camera wearer sliced, diced, and boiled potatoes, interacted with individuals, and modified cooking times. | Incorrect: The camera wearer did not set dice the potatoes. Interacting with individuals is not related to preparation/ cooking of food. |
> | E. The camera wearer peeled, chopped, and sautéed vegetables, interacted with individuals, and adjusted cooking settings, demonstrating a methodical approach to meal preparation. | Incorrect: The camera wearer did not sauté vegetables. Interacting with individuals is not related to preparation/ cooking of food. |
>
> As shown in the example above, the presence of hard negatives ensures that the MCQs are challenging, and a nuanced and detailed understanding of events in the video is required to answer the question correctly.
>
>
>
> $~$
>
> > In line 198 you mention 3 human experts conducted evaluations - what qualifies them as a human "expert"? and what does it mean that they achieve accuracy of 84.4% (is that an average? or did only one human answer each question, something else?)
>
> The evaluations were conducted by the authors. To prevent contamination, we strictly made sure that the evaluators had no prior knowledge of the ground truth answers before assessing the questions. Each multiple-choice question was answered by one human, and the reported accuracy of 84.4% represents the average performance across 96 MCQs.

---

> ### Author Rebuttal · Authors · 2024-08-17
>
> ## Response to Reviewer Z2rz (Part 1/2)
>
> We sincerely thank the reviewer for the comprehensive and encouraging feedback. Below, we address each concern raised by the reviewer.
>
> > Dataset construction procedure is resource-intensive limiting adoption of similar mechanisms by other groups on new datasets
>
> We acknowledge reviewer’s concern regarding the cost and human effort required to construct HourVideo. **This resource-intensive development was necessary to pioneer data-centric progress in long-form video-language understanding**. Furthermore, we remark that with time, the cost of building similar benchmarks is expected to decrease, making them more accessible. In the case of HourVideo, as Multimodal LLMs continue to advance and become more cost-effective, the need for extensive human intervention is expected to decrease, reducing overall costs for future efforts of similar scale. **Our submission includes comprehensive details to ensure reproducibility, enabling the academic community to build upon/ adapt our work.**
>
> $~$
>
> > Benchmark evaluations rely on existing methods/LLMs, would have been interesting to see some novel method applied to this dataset
>
> We appreciate the suggestion. In this study, we have strategically focused on benchmarking existing approaches based on core multimodal capabilities comprehensively. We fully recognize the potential and value of exploring novel methods for tackling HourVideo and consider this an important avenue for future work. The scope of this paper is primarily to establish the benchmark, setting a foundation upon which innovative methods can be developed by the research community in the future.
>
> $~$
>
> > Restricted to Egocentric videos (though to proper audience thats probably a pro lol)
>
> Thank you for the comment. As discussed in the paper (Sec. 2.2), we chose egocentric videos because their perspective closely aligns with typical visual inputs for **autonomous agents/ assistants**. In particular, we selected the Ego4D dataset for its relevance and accessibility: (1) it contains long egocentric videos, (2) it offers extensive visual narrations that enhance the creation of diverse multiple-choice questions, and (3) it is readily accessible under the Ego4D license. Additionally, we avoided using other types of video content such as movies, sports clips, or YouTube videos due to privacy and licensing concerns.
>
> $~$
>
> > I wonder if it would make sense to include benchmark numbers for whatever is SOTA in short form video understanding to demonstrate potential gaps of those models as well
>
> Thank you reviewer for the insightful suggestion. We address reviewer’s comment by conducting experiments using the recently released **Tarsier model which reports state-of-the-art results in multiple short-form video understanding benchmarks**. Following the exact setup in Tarsier for long-video understanding, we use the publicly available Tarsier-7B model with 16 frames uniformly sampled from the entire video. The results for HourVideo benchmark (500 videos) are as follows:
>
> |                                   | **Summarization** | **Perception** | **Visual Reasoning** | **Navigation** | **Average** |
> | --------------------------------- | ----------------- | -------------- | -------------------- | -------------- | ----------- |
> | Blind LLMs                        | 24.3              | 20.0           | 19.2                 | 20.2           | 19.7        |
> | Socratic LLMs: GPT-4             | 41.0              | 29.4           | 23.3                 | 10.9           | 25.8        |
> | Socratic LLMs: LLaVA-NeXT-34B-DPO | 34.5              | 26.8           | 19.2                 | 21.8           | 22.3        |
> | Tarsier-7B (16 frames)            | 32.2              | 24.7           | 27.6                 | 17.9           | 26.9        |
> | **Gemini 1.5 Pro**                | **56.7**          | **40.8**       | **37.8**             | **35.7**       | **39.9**    |
>
> The Tarsier-7B performs similar to Socratic LLMs. These additional results will be included in the final version.
>
> $~$
>
> (To be continued)

---

> ### Author Response · Authors · 2024-08-29
> **Seeking Feedback from Reviewer Z2rz**
>
> Dear Reviewer **Z2rz**,
>
> With the author-reviewer discussion phase concluding in just over three days, we would greatly appreciate it if you could let us know whether our responses are satisfactory.
>
> We have addressed all your concerns and conducted the suggested experiments using the Tarsier-7B model. These results will be included in the final version. We are happy to address any further concerns you may have.
>
> We sincerely appreciate your time and effort.
>
> Thank you,
>
> Authors

---

> > ### Author Response · Authors · 2024-09-01
> > **Thank you Reviewer Z2rz for the encouraging feedback**
> >
> > Dear Reviewer **Z2rz**,
> >
> > We thank the reviewer for the comprehensive and encouraging feedback.
> >
> > As today is the final day of the discussion phase, we would appreciate it if you could let us know whether our responses are adequate. We’re happy to provide any further information.
> >
> > Thank you for your time.
> >
> > Best,
> >
> >  Authors

---

> > > ### Comment · Reviewer_Z2rz · 2024-09-01
> > > **response**
> > >
> > > looks good / thanks for responses. best of luck with rest of process

---

> > ### Author Response · Authors · 2024-09-01
> > **Thank you for your response**
> >
> > We sincerely thank the reviewer for the feedback!
> >
> > Best,
> >
> > Authors

---

### Official Review · Reviewer_rfPC · 2024-07-25
**A large-scale long-form MC VQA dataset**

**Rating:** 6
**Confidence:** 3
**Correctness:** Yes.
**Clarity:** Yes.

**Review:**

This large HourVideo dataset is created based on a clear data taxonomy. The task suite includes  comprehensive question types. Also, the authors develop a LLM-human hybrid pipeline to generate the data in order to keep balance between automation (efficiency) and quality. The introduction to the taxonomy and the pipeline is clear and easy to understand. Of course this is one of the most large-size MC VQA dataset, however, due to its scale, the cost of the dataset building is pretty high (>\$10k + hundreds of expert human effort). Also, the videos are borrowed from the existing Ego4D dataset, where may have the potential data leakage problem.

Another issue is that, although this dataset is large, it cannot be used to train models because it contains only 500 long videos. The length of these videos also makes evaluation extremely costly (e.g., evaluating Gemini Pro on 276 videos costs $105 per one-hour video). Additionally, the vast majority of multi-modal models do not support long videos and texts directly, which undermines the dataset's current significance. However, it may become a useful benchmark in the future.

There are also other issues, such as the dataset's content (only containing ego-centric data) and potential biases, which are acknowledged in the limitations section.


Pros:
1. The largest MC VQA dataset;
2. Comprehensive question types and a LLM-human data generation pipeline;
3. The benchmarking results illustrate the significant performance gap between humans and models, highlighting areas for improvement.

Cons:
1. Requires significant human effort to build the dataset;
2. Not a suitable or low-cost benchmark for most current multi-modal models due to the lengthy videos and high evaluation costs;
3. Dataset contains only ego-centric data from existing dataset, leading to potential biases and data leakage.

**Strengths:**

See Pros in Review

**Additional Feedback:**

Could you please emphasize the necessity of building a long-form MC VQA benchmark? It appears that even humans struggle to answer detailed questions after watching such lengthy videos only once. The long-form video multiple-choice tasks do not seem to test the reasoning and understanding capabilities of models. Instead, these tasks focus more on the capability of temporal retrieval, which could be solved in a relatively simple manner but not through a LMM by feeding the whole videos.

**Documentation:**

Yes.

**Ethics:**

No.

**Limitations:**

Yes.

**Opportunities For Improvement:**

See Cons in Review

**Relation To Prior Work:**

Yes.

**Summary And Contributions:**

In this paper, the authors create a new MC VQA evaluation dataset based on the Ego4D dataset, which includes 381 hours of long-form videos. They test several baseline models on this dataset, revealing a significant gap compared to human performance.

---

> ### Author Rebuttal · Authors · 2024-08-17
>
> ## Response to Reviewer rfPC
>
> We sincerely thank the reviewer for their valuable time.
> > Requires significant human effort to build the dataset.
>
> We acknowledge the reviewer's concern regarding the significant human effort required to construct HourVideo. However, as Multimodal Large Language Models (LLMs) continue to advance and become more cost-effective, we anticipate a decrease in the need for extensive human intervention, thereby reducing overall costs for future efforts of similar scale.
>
> **It's important to note that creating datasets at the frontier of model capabilities is inherently costly**. Yet, as model capabilities improve and inference becomes more affordable, we can generate larger and higher-quality datasets. This trend is evident in the progression from COCO 2014 (1.2M masks) to SAM 2023 (1B masks). We're already observing a similar trend in LLMs, where the inference cost of GPT-4o is now six times lower than that of GPT-4. This trajectory suggests that future iterations of multimodal models will likely enable more efficient and cost-effective dataset creation similar to HourVideo.
>
> $~$
> > Not a suitable or low-cost benchmark for most current multi-modal models due to the lengthy videos and high evaluation costs
>
> Thank you for your comment. While we acknowledge the challenges posed by long videos and high evaluation costs, HourVideo is **intentionally** designed to push the boundaries of long-form video understanding capabilities of models. Our goal is not simply to assess models on tasks within their current reach but to catalyze the development of models capable of understanding long-form videos that mirror **human-level intelligence** in complex, realistic scenarios. Furthermore, evaluation costs are continuously decreasing. For instance, GPT/ Gemini inference costs have reduced by more than 50% this year.
>
> $~$
> > Could you please emphasize the necessity of building a long-form MC VQA benchmark?
>
> Our world presents an endless stream of visual stimuli. Humans demonstrate a remarkable ability to process such stimuli over long time horizons, plan and act based on complex, continuous inputs. HourVideo benchmark is inspired by this fundamental human capability, aiming to pioneer efforts on one-hour video understanding. **We believe progress in HourVideo will pave way for developing applications that require sustained visual processing, such as augmented reality assistants and embodied agents.** Our Broader Impact section (Supplementary - D) discusses these applications in detail. We also remark that recent advancements in multimodal models are closing the gap between AI and human performance in understanding videos up to 3 minutes. For example, Gemini 1.5 Pro achieves 72.2% accuracy on EgoSchema [f], approaching human performance of 76.2%.
>
> [f] Egoschema: A diagnostic benchmark for very long-form video language understanding. *NeurIPS 2024*
>
> $~$
> > Although this dataset is large, it cannot be used to train models because it contains only 500 long videos.
>
> We agree. HourVideo is a **benchmark** **dataset** constructed to **evaluate** long-form video-language understanding. We don’t expect the community to train models on HourVideo, and will make it more explicit in the paper.
>
> $~$
> > Dataset contains only ego-centric data from existing dataset, leading to potential biases and data leakage.
>
> - **Justification for choosing egocentric videos from Ego4D:** As discussed in Sec. 2.2, we chose egocentric videos because their perspective closely aligns with typical visual inputs for **autonomous agents/ assistants**. We chose the Ego4D dataset due to its relevance and accessibility: it contains long egocentric videos, provides extensive visual narrations for diverse MCQ creation, and is accessible under the Ego4D license. Additionally, we avoided using other types of videos such as movies, sports clips, or YouTube videos due to privacy and licensing concerns. Our priority is to ensure the dataset remains readily available for future research, free from scraping or copyright complications.
>
> - **Dataset biases**: Although the curators of the Ego4D dataset have noted certain biases (e.g., biases towards urban or college town areas, at-home scenarios, and a lack of major social events), the videos in Ego4D are impressively diverse, spanning 74 cities in 9 countries, 5 continents and 931 unique camera wearers of varying ages, nationalities, and gender identities. By leveraging Ego4D, we significantly improve our ability to accurately benchmark long form video understanding across a wide range of actors and environments.
>
> - **Data leakage**: To address concerns regarding data leakage, our leaderboard will include an option for participants to disclose whether their models were trained on Ego4D videos. This will ensure fair comparisons.
>
> $~$
> > The long-form video multiple-choice tasks do not seem to test the reasoning and understanding capabilities of models. Instead, these tasks focus more on the capability of temporal retrieval, which could be solved in a relatively simple manner but not through a LMM by feeding the whole videos.
>
> We respectfully disagree with this comment. While temporal localization is a necessary skill, it alone is insufficient to solve all the complex tasks presented in HourVideo. In particular, tasks including perception (tracking), visual reasoning (spatial, temporal, predictive, causal, counterfactual), and navigation (room-to-room, object retrieval) in HourVideo necessitate a deeper understanding, **requiring models to not only localize but also** **synthesize, connect and reason across disparate video segments for extended durations.** The above tasks correspond to 66.7% of MCQs in HourVideo. For example, consider navigation questions in HourVideo. It demands models to do more than merely identify specific moments; models must construct mental models of the physical world over long horizons to accurately answer MCQs. We show multiple such examples in Figure 1.

---

> > ### Comment · Reviewer_rfPC · 2024-08-23
> > **Response to Authors**
> >
> > I appreciate your response. I think the authors resolved some of my concerns. I would like to raise my score to 6.

---

> > ### Author Response · Authors · 2024-08-24
> > **Thank you Reviewer rfPC for increasing your rating**
> >
> > Thank you for reviewing our rebuttal and increasing your rating to 6. We are happy to address any further concerns you may have. We sincerely appreciate your time and effort.
> >
> > Thank you,
> >
> > Authors

---

### Official Review · Reviewer_QtgE · 2024-08-08
**Largest video-language understanding dataset**

**Rating:** 9
**Confidence:** 4
**Correctness:** The claims in this paper needs to be …
**Clarity:** yes

**Review:**

The paper's contribution is significant and timely. There is lack of public long video-language understanding dataset and authors have tried to fill this gap with HourVideo. Moreover, the evaluation of several multimodal models show that long term video comprehension is still a difficult task for current models.

**Strengths:**

1. Authors create a long-form video-language understanding dataset with 17 tasks.
2. Dataset generation pipeline is carefully designed that results in creation of difficult dataset. This pipeline results in accurately measuring the video-language understanding of multimodal models. Specifically, the blind filtering and manual question generation is helpful in increasing the quality of the dataset.
3. Evaluation of multimodal models shows that the created dataset is difficult. Moreover, the performance difference between Socratic models and extreme long context multimodal models suggests that long-term video compression is required to perform better on HourVideo dataset.

**Additional Feedback:**

See the comments above.

------------

Authors have addressed my concerns in the rebuttal. Hence, I have increased the paper rating.

**Documentation:**

yes

**Ethics:**

No.

**Limitations:**

Limitations are discussed.

**Opportunities For Improvement:**

1.  Is the comparison statement in the abstract  scientifically correct?  Statement: "human baselines significantly outperform the state-of-the-art long-context multimodal model Gemini Pro 1.5 (84% vs. 40%)" . In the main text, authors report humans were evaluated on only 96 MCQs, while Gemini Pro 1.5 seems to be evaluated on 13,003 questions resulting in 40% average performance. Is this a fair comparison?
2. In Table 3,  minimum duration of HourVideo dataset is reported to 45.7 while in section 2.3, average duration of HourVideo dataset is reported to 45.7. Please clarify.
3. Related work [1, 2] should be discussed in the paper.

References
1. Yang, Xitong, et al. "Relational space-time query in long-form videos." Proceedings of the IEEE/CVF Conference on Computer Vision and Pattern Recognition. 2023.
2. Strafforello, Ombretta, Klamer Schutte, and Jan Van Gemert. "Are current long-term video understanding datasets long-term?." Proceedings of the IEEE/CVF International Conference on Computer Vision. 2023.

**Relation To Prior Work:**

Authors should discuss few related works. Please refer weakness section.

**Summary And Contributions:**

The authors release an egocentric long-form video dataset, HourVideo with the focus on video-language understanding. The evaluation of long form video understanding models through tasks is a difficult problem and requires designing questions in the task suite that cannot be easily answered through short clips. In HourVideo, authors introduce several challenging tasks such as summarization, perception, visual reasoning and navigation that requires long-term comprehension of the video. The dataset generation pipeline outlines the question generation process and multiple filtering stages. The resultant dataset is the biggest publicly available dataset with 500 videos and 13k+ questions across 17 tasks.

---

> ### Author Rebuttal · Authors · 2024-08-17
>
> ## Response to Reviewer QtgE
>
> We sincerely thank the reviewer for the encouraging feedback. Below, we address each concern raised by the reviewer.
>
> > Is the comparison statement in the abstract scientifically correct? Statement: "human baselines significantly outperform the state-of-the-art long-context multimodal model Gemini Pro 1.5 (84% vs. 40%)". In the main text, authors report humans were evaluated on only 96 MCQs, while Gemini Pro 1.5 seems to be evaluated on 13,003 questions resulting in 40% average performance. Is this a fair comparison?
>
> We agree that the numbers presented—84% (96 MCQs) for humans vs. 40% (6221 MCQs) for Gemini Pro 1.5—are not directly comparable. This stems from the substantial costs associated with hiring trained humans, which could **exceed $15,000** to evaluate all 500 videos. To address reviewer’s concern, **we expanded human evaluation to include an additional 7.42 hours of video content for this rebuttal, covering 10.20 hrs/ 206 MCQs/ 13 videos resulting in 82.5% human performance**. We aim to continue expanding these efforts as resources permit.
>
>
>
> $~$
>
> > In Table 3, minimum duration of HourVideo dataset is reported to 45.7 while in section 2.3, average duration of HourVideo dataset is reported to 45.7. Please clarify.
>
> Thank you for the comment, it is a typo. Table 3 reports **average** video duration in seconds. We will fix the title of column 3 in the final version.
>
>
>
> $~$
>
> > Related work [1, 2] should be discussed in the paper.
>
> Thank you for pointing out these works, we will discuss these works in Section 4.

---

> ### Author Response · Authors · 2024-08-21
> **Thank you Reviewer QtgE for increasing your rating to 9**
>
> Thank you for reviewing our rebuttal and increasing your rating to 9. We sincerely appreciate your time and effort.
>
> Thank you,
>
> Authors

---

### Author Rebuttal · Authors · 2024-08-17

## Thank you for the positive and constructive comments


We thank the AC and reviewers for their valuable time and feedback. We appreciate the reviewers' recognition of the **comprehensive** and **challenging** nature of our proposed **HourVideo** **benchmark**. HourVideo is described as the **largest** (R1-QtgE, R2-rfPC), **comprehensive** (R2-rfPC, R3-Z2rz, R4-Qx1u), **thoughtfully-constructed** (R3-Z2rz), **diverse** (R3-Z2rz, R4-Qx1u), and **high-quality** (R3-Z2rz, R4-Qx1u) benchmark. In particular, R1-QtgE highlights that our dataset is **significant** **and timely**. R2-rfPC highlights the **significant performance gap between humans and models on HourVideo**. R3-Z2rz points out that the paper has **potential to spur new research directions / advances in the field**. Lastly, R4-Qx1u appreciates that our paper addresses **a critical gap in multimodal model capabilities and has the potential for significant real-world impact**.


We conducted 3 additional sets of experiments requested by reviewers.

- **Tarsier-7B (R3-Z2rz, R4-Qx1u)**: Following the exact setup in Tarsier for long-video understanding, we use publicly available Tarsier-7B model with 16 frames uniformly sampled from the entire video. Tarsier-7B performs similar to Socratic LLMs suggesting significant research gap.
- **MovieChat (R4-Qx1u):** We benchmark MovieChat on a subset of HourVideo covering 57.0 hours of duration/1533 MCQ tests/ 75 randomly selected videos. MovieChat performs marginally better than random chance highlighting research gap.
- **Ablation study for task-level vs. individual QA evaluation (R4-Qx1u):** We conducted an ablation study covering 15.9 hours of duration/570 MCQ tests across 25 randomly selected videos, where each multiple-choice question (MCQ) was evaluated independently. Our results emphasize the efficiency and validity of our proposed task-level/subtask level evaluation method.

In what follows, we provide comprehensive responses to all questions. We hope that our responses can address the concerns and we hope that reviewers may consider increasing the ratings if our responses are satisfactory.

$~$

Thank you.

Authors

---

### Decision · Program_Chairs · 2024-09-26

**Decision:**

Accept (Poster)

**Comment:**

The paper introduces HourVideo, a benchmark for long, egocentric video understanding which includes 17 sub-tasks formulated as multiple choice question and answering. Most reviewers agree the proposed dataset is challenging, experiments comprehensive, set of tasks are diverse, annotation quality is high which is the results of the carefully-designed dataset generation pipeline. Meta-reviewer agrees with the reviewers and recommend to accept the paper.